# Cofactors facilitate *bona fide* prion misfolding *in vitro* but are not necessary for the infectivity of recombinant murine prions

**Miguel Ángel Pérez-Castro**[1☉], **Hasier Eraña**[1,2,3☉], **Enric Vidal**[4,5], **Jorge M. Charco**[1,2,3],
**Nuria L. Lorenzo**[6], **Nuno Gonçalves-Anjo**[1], **Josu Galarza-Ahumada**[1],
**Carlos M. Díaz-Domínguez**[1], **Patricia Piñeiro**[1], **Ezequiel González-Miranda**[1],
**Samanta Giler**[4,5], **Glenn Telling**[7], **Manuel A. Sánchez-Martín**[8,9], **Joseba Garrido**[10],
**Mariví Geijo**[10], **Jesús R. Requena**[6], **Joaquín Castilla** [1,2,11]*

**1** Center for Cooperative Research in Biosciences (CIC BioGUNE), Basque Research and Technology Alliance (BRTA), Derio, Spain, **2** Centro de Investigación Biomédica en Red de Enfermedades infecciosas (CIBERINFEC), Carlos III National Health Institute, Madrid, Spain, **3** ATLAS Molecular Pharma S. L., Derio, Spain, **4** IRTA, Programa de Sanitat Animal, Centre de Recerca en Sanitat Animal (CReSA), Campus de la Universitat Autònoma de Barcelona (UAB), Bellaterra, Catalonia, Spain, **5** Unitat mixta d'Investigació IRTA-UAB en Sanitat Animal, Centre de Recerca en Sanitat Animal (CReSA), Campus de la Universitat Autònoma de Barcelona (UAB), Bellaterra, Catalonia, Spain, **6** Department of Medical Sciences, CIMUS Biomedical Research Institute, University of Santiago de Compostela-IDIS, Santiago de Compostela, Spain, **7** Prion Research Center, Colorado State University, Fort Collins, Colorado, United States of America, **8** Department of Medicine, Transgenic Facility, University of Salamanca, Salamanca, Spain, **9** Institute of Biomedical Research of Salamanca (IBSAL), Salamanca, Spain, **10** Animal Health Department, NEIKER-Basque Institute for Agricultural Research and Development, Basque Research and Technology Alliance (BRTA), Derio, Spain, **11** IKERBASQUE, Basque Foundation for Science, Bilbao, Spain

☉ These authors contributed equally to this work.
* jcastilla@cicbiogune.es

## Abstract

Prion diseases, particularly sporadic cases, pose a challenge due to their complex nature and heterogeneity. The underlying mechanism of the spontaneous conversion from PrP^C to PrP^Sc, the hallmark of prion diseases, remains elusive. To shed light on this process and the involvement of cofactors, we have developed an *in vitro* system that faithfully mimics spontaneous prion misfolding using minimal components. By employing this PMSA methodology and introducing an isoleucine residue at position 108 in mouse PrP, we successfully generated recombinant murine prion strains with distinct biochemical and biological properties. Our study aimed to explore the influence of a polyanionic cofactor in modulating strain selection and infectivity in *de novo*-generated synthetic prions. These results not only validate PMSA as a robust method for generating diverse *bona fide* recombinant prions but also emphasize the significance of cofactors in shaping specific prion conformers capable of crossing species barriers. Interestingly, once these conformers are established, our findings suggest that cofactors are not necessary for their infectivity. This research provides valuable insights into the propagation and maintenance of the pathobiological features of cross-species transmissible recombinant murine prion and highlights the intricate interplay between cofactors and prion strain characteristics.

**Data availability statement:** All the data related to the manuscript is included in the main text or in Supporting information file. Raw data used for calculations and plots is publicly available in the ZENODO repository with doi: https://doi.org/10.5281/zenodo.14360226. Data can be accessed through the following link: https://zenodo.org/records/14360226.

**Funding:** The present work was partially funded by three different grants awarded by "Agencia Estatal de Investigación, Ministerio de Ciencia e Innovación" (Spanish Government), grant numbers PID2021-122201OB-C21 granted to J.C., PID2021-1222010B-C22 granted to E.V., and PID2020-117465GB-I00 granted to J.R.R., funded by MCIN/ AEI /10.13039/501100011033 and co-financed by the European Regional Development Fund (ERDF). EFA031/01 NEURO-COOP, which is co-funded at 65% by the European Union through Programa Interreg VI-A España-Francia-Andorra (POCTEFA 2021-2027) granted to J.C. and H.E. Additional funding was provided by the Instituto de Salud Carlos III (ISCIII), grant number AC21_2/00024, granted to J.C. Additionally, CIC bioGUNE currently holds a Severo Ochoa Excellence accreditation, CEX2021-001136-S granted to J.C., also funded by Ministerio de Ciencia e Innovación/ AEI /10.13039/501100011033. Transgenic Facility, directed by M.A.S.-M., is supported by Instituto de Salud Carlos III (ISCIII), co-funded by the European Union grant PT23/00123. The funders had no role in study design, data collection and analysis, decision to publish, or preparation of the manuscript.

**Competing interests:** I have read the journal's policy and the authors of this manuscript have the following competing interests: Authors HE and JMC are employed by the commercial company ATLAS Molecular Pharma SL. This does not alter their adherence to all Journal policies on sharing data and materials and did not influence in any way the work reported in this manuscript. The rest of the authors declare no competing interests.

## Author summary

Prion diseases are rare and complex neurodegenerative disorders that can occur spontaneously, through a poorly understood conformational or structural change of normal, physiological prion protein. This abnormally structured form, known as prion, acquires the capacity to induce the same transformation to surrounding prion proteins, leading to disease. In our study, we take advantage of a recently developed methodology that closely mimics this process in a test tube using extremely simple components. Applying this system and modifying a specific part of the mouse prion protein, we were able to generate different prion strains with unique characteristics spontaneously. This process is greatly enhanced using an additional molecule called cofactor, that has been proposed to affect the variability and infectivity (capacity to cause disease in an animal model) of these prions. Our findings show that while cofactors facilitate the spontaneous formation of prions and may influence their final characteristics, they are not necessary for their infectivity nor for their spontaneous formation. This research gives new insights into the role cofactors play in the spontaneous generation of prions and their behavior in animal models.

## Introduction

Prion diseases are a group of ravaging neurodegenerative disorders that affect a wide range of mammals including humans. These pathologies are caused by the conformational conversion of the cellular prion protein (PrP$^C$) into a pathological conformer (PrP$^{Sc}$) [1]. These two iso-forms are distinguished by their three-dimensional structures, being the pathological isoform neurotoxic and able to induce its conformation to the natively folded cellular counterpart [2]. Regarding its origin, this conformational conversion can be related to the presence of specific *PRNP* mutations (genetic prion diseases), induced by previously formed PrP$^{Sc}$ acquired from exogenous sources such as iatrogenic or accidental transmission (acquired prion diseases), and putatively spontaneous, being the most common aetiology in humans (sporadic or idiopathic prion diseases) [3]. However, the causes that trigger the supposedly spontaneous misfolding of the PrP$^C$ are completely unknown, despite their relevance to disease prevention and the potential opportunities for developing prophylactic treatments.

After decades of research on prion diseases in several different mammalian species, we now understand that the prion protein can misfold into multiple pathogenic conformations [4–9], further complicating the study of spontaneous misfolding events. These conformers can specifically propagate their structure to the native counterpart and exhibit distinctive biochemical and biological properties, giving rise to what the prion research field terms as prion strains [10]. The existence of multiple prion strains for a single PrP sequence explains the highly variable and clinically heterogeneous course of sporadic prion diseases [11]. This phenomenon, including the possibility of multiple strains coexisting within the same host, is not exclusive to human prion diseases. It has been reported in most mammals in which prion diseases have been described, such as sheep and moose [12,13]. It has been studied in laboratory models of prion disease, primarily in rodents, to which multiple strains from different species have been transmitted. In some cases, these transmitted strains retain the distinctive characteristics of the original inoculum, while in others, they evolve into new strains with properties distinguishable from other conformers [14]. Despite the knowledge gathered about prion strains over these decades, the molecular mechanisms determining if a prion protein will arrange into one conformer or another upon misfolding remain unknown. It is also

unclear whether the tertiary/quaternary arrangement of a given prion strain is conditioned by any PrP-independent factors (known as cofactors) or could be completely random, depending solely on the first stable PrP$^{Sc}$ propagation nuclei formed.

Research on the spontaneous PrP$^C$ misfolding and prion strain determinants has long been hindered by the scarcity and unpredictability of sporadic prion disease cases in both natural and laboratory animal models. Over the past three decades, numerous transgenic murine models have been generated to address this situation [15]. However, animal models of sporadic prion disease mostly relied on *PRNP* overexpression or the expression of PrP variants with pathology-associated mutations that trigger the conversion of PrP$^C$ into PrP$^{Sc}$ [16,17]. These models poorly mimic the spontaneous misfolding events occurring in nature and are too complex to address questions about potential cofactors influencing different strain formation. An exception is found in transgenic mice expressing wild-type prion protein from bank voles (*Myodes glareolus*). Animals expressing the polymorphic variant with isoleucine at position 109 develop a spontaneous prion disease when overexpressed, unlike those expressing the methionine variant at the same position [18]. Additionally, the development of cell-free or *in vitro* prion propagation systems has provided new opportunities to study the mechanisms involved in spontaneous misfolding in the absence of pathology-associated genetic alterations. These systems have solved the problem of low incidence in *in vivo* models and provided simplified platforms to investigate the molecular determinants of prion strains [19]. The first of such systems led in fact to the spontaneous generation of *bona fide* synthetic prions [20] and demonstrated that different conformers could be generated *in vitro*, showing differential strain features that caused diverse disease phenotypes [21–25], all prepared in the presence of distinct concentrations of denaturing agents such as urea or guanidine hydrochloride.

Grounded on these initial cell-free prion propagation systems or amyloid seeding assays [26,27], further developments led to one of the most widely used tools in this regard, the Protein Misfolding Cyclic Amplification (PMCA), developed over twenty years ago [28]. This system allowed, among other things, the generation of spontaneously misfolded PrP from brain homogenates of wild-type animals used as substrate without the need of using denaturing conditions during the reaction. These prions generated spontaneously *in vitro* in native conditions, while infectious *in vivo*, showed an unusually long incubation period and displayed different strain properties compared to previously characterized prion strains [29,30]. However, further developments in this method, primarily through the use of recombinant PrP produced in *Escherichia coli* and chemically defined substrates devoid of brain homogenate, led to an important milestone in prion disease research: the spontaneous generation of a highly infectious recombinant prion in PMCA using minimal components [31]. From a PMCA substrate containing only recombinant mouse PrP, RNA and POPG (1-Palmitoyl-2-oleoyl-sn-glycero-3-phosphoglycerol), these synthetic prions were able to induce prion disease in a wild-type model, showing incubation periods similar to those obtained with classical prion strains and dispelling any doubts about the proteinaceous nature of the prion agent [32]. Although this study represented a breakthrough in prion research, subsequent studies attempting to reproduce the spontaneous generation of synthetic prions by PMCA using minimal components revealed difficulties and resulted in non-infectious preparations [33]. Even the authors of the original publication found non-infectious products and heterogeneous recombinant prion mixtures [34,35], likely related to both the methodology and the minimal components used.

Beyond promoting *bona fide* prion misfolding *in vitro*, the role of RNA - the first misfolding-inducing cofactor described [36] - or that of other polyanionic or lipidic cofactors on the characteristics of the generated misfolded products is still unclear. While some studies indicate that cofactors such as distinct polyanions and lipids could be essential for prion infectivity [31,37,38], other studies propose a role in determining specific prion

conformer or strain formation, which in turn would determine the infectious capacity of each specific conformer [39–41]. This latter hypothesis is also supported by the variable outcome in terms of infectivity mentioned previously for synthetic prions generated in the presence of RNA and POPG, which may allow some degree of strain variability, and the generation of recombinant infectious prions in cofactor-free environment, although this approach required seeding with previously formed prions or the use of denaturing agents [25,42,43]. Nonetheless, the lack of highly infectious synthetic prions generated spontaneously in the absence of cofactors and the difficulties in determining differences in the biological and biochemical features of synthetic prions hinder obtaining robust conclusions on their role in infectious prion formation.

In this work, we explore the role of a polyanionic cofactor in spontaneous prion generation using Protein Misfolding Shaking Amplification (PMSA). PMSA is a recently developed variant derived from the previous amyloid seeding assays based on the use of recombinant PrP [26,27] that overcomes some variability issues of PMCA, allowing consistent and swift generation of hundreds of infectious recombinant prions using recombinant PrP and dextran sulfate as a polyanionic cofactor [44,45]. Here, we generated distinct recombinant murine prions spontaneously by substituting a single residue in mouse PrP (L108I, mimicking the 109I polymorphism from bank vole PrP) and using dextran sulfate and glass beads to promote spontaneous prion misfolding *in vitro*. This substitution, as expected from previous studies with mice expressing bank vole PrP variants [18], increased the spontaneous misfolding proneness of the murine protein. Our approach rapidly yielded *bona fide* synthetic murine prion strains with differential properties *in vivo*. By propagating some of these strains in the absence of cofactors, we demonstrated that while cofactors are key in facilitating spontaneous misfolding, they are not necessary for the infectivity of the recombinant preparations. Our findings provide evidence that cofactors act as modulators or selectors of different conformers, influencing the strain properties of synthetic prions. Additionally, we show that recombinant infectious prions can be generated *de novo* in the absence of dextran sulfate. This further confirms that cofactors are not necessary for prion infectivity but may influence the biological and biochemical properties of prions, likely in a strain-dependent manner.

## Materials and methods

### Ethics statement

TgVole (1x) (FVB/N.129Ola-Tg(Prnp-Bvole109I)C594PRC/Cicb) and TgMoL108I (3x) (B6&CBA.129Ola-Tg(Prnp-Mo108I)1Sala/Cicb) founder mice were generated at Transgenic Facility of the University of Salamanca (Spain) and breeding colonies were kept on a 12:12 light/dark cycle, receiving food and water *ad libitum* at Asociación Centro de Investigación Cooperativa en Biociencias - CIC bioGUNE - (Spain). C57BL/6 mice were bred in-house in the same conditions at the Center for Experimental Biomedicine (CEBEGA - University of Santiago de Compostela, Spain). All animals were inoculated at the Center for Experimental Biomedicine (CEBEGA - University of Santiago de Compostela) and Neiker - Basque Institute for Agricultural Research and Development. All experiments involving animals in Spain adhered to the guidelines included in the Spanish law "Real Decreto 53/2013 de 1 de febrero" on protection of animals used for experimentation and other scientific purposes, which is based on the European Directive 2010/63/UE on Laboratory Animal Protection. The project was approved by the Ethical Committees on Animal Welfare (project codes assigned by the Ethical Committee CIC bioGUNE-2023/23, NEIKER-2021/49, and CEBEGA-15012/2023/002) and performed under their supervision.

## Expression and purification of recombinant PrP

Bacterial expression and purification of different murine recombinant PrP (amino acids 23-231) (rec-PrP) were performed as previously described [46]. Briefly, the pOPIN E expression vector containing the wild-type murine *Prnp* gene was prepared using standard molecular biology techniques. The oligonucleotides 5' AGGAGATATACCATGAAAAAGC-GGCCAAAGCCTGAA 3' and 5' GTGATGGTGATGTTAGGATCTTCTCCCGTCGTAATA 3' (Sigma-Aldrich) were used to obtain the PrP of interest from genomic DNA of wild-type mice (C57BL/6 genetic background). The In-Fusion cloning kit (Clontech) was used for its introduction into the pOPIN E expression vector (Oxford Protein Production Facility, OPPF). From this plasmid, 19 additional pOPIN E expression vectors containing murine *Prnp* genes with substitutions at position 108 for the 19 different naturally occurring amino acids were generated. For this, a two-step PCR site-directed mutagenesis was performed using 19 pairs of oligonucleotides (S1 Table) and two additional oligonucleotides 5' CCGCGGGGGGACG-GCTGCC 3' and 5' GAACAGAGGTGCGTCTGGTG 3' (Sigma-Aldrich) that hybridize with a sequence of the mouse pOPIN E PrP$_{23-231}$ vector.

*E. coli* Rosetta (DE3) competent cells (EMD Millipore) were transformed with each expression vector using a standard heat-shock transformation procedure [45], allowing the expression of the 20 recombinant proteins. Transformed bacteria were cultured in LB broth (Pronadisa) with 50 µg/ml of Ampicillin sodium salt (Sigma-Aldrich) at 37 °C with vigorous agitation. Recombinant protein expression was induced by the addition of Isopropyl β-D-1-thiogalactopyranoside (IPTG) (Gold biotechnology) at 1 mM for 3 hours. After induction, bacterial cultures were centrifuged at 4,500 *g* for 15 min at 4 °C. The bacterial pellets were resuspended in 50 ml of lysis buffer containing 50 mM Tris-HCl (Fisher Bioreagents), 5 mM EDTA (Sigma-Aldrich), 1% Triton X-100 (Sigma-Aldrich), 1 mM PMSF (Sigma-Aldrich), adjusted to pH 8, and 100 µg/ml lysozyme (Sigma-Aldrich). The suspension was then incubated with 100 U/ml DNase (Sigma-Aldrich) and 20 mM MgCl$_2$ (Sigma-Aldrich) for 30 min with shaking at 200 rpm at room temperature. The lysates were subsequently centrifuged at 8,500 *g* and 4 °C for 1 hour. The resulting pellets, containing bacterial inclusion bodies, were resuspended in 50 ml of washing buffer [20 mM Tris-HCl, 150 mM NaCl (Fisher Chemicals), 1 mM EDTA, 1% Sarkosyl (Sigma-Aldrich), pH 8]. After an additional centrifugation at 8,500 *g*, 4 °C for 1 h, the new pellet was dissolved in inclusion buffer [20 mM Tris-HCl, 0.5 M NaCl and 6 M GdnHCl (Sigma-Aldrich), pH 8] and incubated overnight at 37 °C with vigorous shaking to break down inclusion bodies and solubilize the recombinant protein. A final centrifugation at 8,500 *g* and 4 °C for 1 h and filtration thought a 0.20 µm-pore syringe filters (Corning) were performed prior to purification. Purification was carried out using a histidine affinity column (HisTrap FF Crude 5 ml chromatographic columns, GE healthcare) coupled to an Äkta Start FPLC equipment (GE Healthcare), taking advantage of the natural His residues present in the octapeptide repeat region of the PrP. After washing with binding buffer [20 mM Tris-HCl, 500 mM NaCl, 2 M GdnHCl and 5 mM Imidazole (Sigma-Aldrich), pH 8], the proteins were eluted in 30 ml of elution buffer containing 20 mM Tris-HCl, 500 mM NaCl, 500 mM imidazole, and 2 M GdnHCl, pH 8. The quality and purity of protein batches was assessed by BlueSafe (NZYtech) staining after electrophoresis in SDS-PAGE gels (BioRad). Finally, GdnHCl was added to a final concentration of 6 M. The final concentration of the protein was adjusted to 25 mg/ml by concentrating the eluted solution using 10 kDa centrifugal filter units (Amicon ultra-15, 10 kDa, Millipore). The batches were then aliquoted and stored at −80 °C until use.

## Preparation of PMSA substrates

For PMSA substrate preparation, all proteins were thawed, diluted 1:5 in PBS (HyClone), and dialyzed against PBS (1:10,000 dilution) for 1 h to allow refolding to their native

conformation. After centrifugation at 19,000 $g$ for 15 min at 4 °C, the supernatants containing dialyzed proteins were further diluted 1:10 in conversion buffer (CB) (0.15 M NaCl and 1% Triton-X-100 in PBS). Two different recombinant PMSA substrates were prepared: those complemented with 0.5% dextran sulfate 6,500–10,000 kDa (Sigma-Aldrich) (also referred as Dx or dex in some figures and text) and those prepared without the addition of any specific cofactor (designated as CB, since they contain just the conversion buffer which include 0.15 M of NaCl and 1% Triton-X-100 and recombinant protein). For comparative PMSA assays involving different protein-containing substrates, rec-PrP concentrations were equaled after dialysis using the Bicinchoninic acid (BCA) assay (ThermoFisher Scientific). In these cases, final protein concentration within each substrate was additionally verified by electrophoresis and total protein staining with BlueSafe (Nzytech), as detailed below.

## Generation of spontaneous recombinant prions by PMSA

Spontaneous generation of recombinant *bona fide* prions was performed using PMSA as described previously [45]. Briefly, 200 mg of acid-washed 0.1 mm glass beads (Sigma-Aldrich) were placed in clean, labeled 2-ml tubes with screw caps (referred as big tubes, bt), containing 500–800 μl of fresh substrate or alternatively, only in some of the initial spontaneous misfolding experiments in labelled 0.2-ml tubes (referred as small tubes, st), containing 50–80 μl of fresh substrate. PMSA was performed for 24 h per round (unless otherwise indicated for specific experimental settings with shorter reaction times), 39 °C and with continuous shaking at 700 rpm using programmable thermoblocks (Digital Shaking Drybath, Thermo Scientific). When more than one round was required for spontaneous misfolding or to propagate spontaneously generated PMSA products, serial PMSA propagation rounds were performed. For these subsequent rounds, tubes with fresh substrate were complemented with 1 mm zirconia-silicate beads (BioSpec Products Inc.) and seeded with the PMSA product from the previous round at 1:10 or 1:100 dilution. After a 24-hour PMSA round under the same conditions described above, the product of this second PMSA round was used as seed at 1:10 to 1:100 dilution for a third round of PMSA. This process was repeated as many times as required and specified for each experiment in the results section.

## Detection of misfolded, proteinase K-resistant, recombinant PrP (rec-PrP$^{res}$) by total protein staining

The PMSA products were digested with proteinase K (PK) and subjected to electrophoresis and total protein staining to assess the presence of PK-resistant misfolded PrP (rec-PrP$^{res}$) in each sample. Briefly, 400 μl of each PMSA product were transferred to 1.5-ml Eppendorf tubes, and PK (Roche) was added to a final concentration of 25 μg/ml. The tubes were incubated at 42 °C for 1 h in a laboratory oven (Nahita) and immediately centrifuged at 19,000 $g$ and 4 °C for 15 min. The supernatant was carefully discarded, and the pellet was washed with 500 μl of PBS and centrifuged again at 19,000 $g$ for 5 min at 4 °C. After removing the PBS, the pellet was resuspended in 15 μl of 1x loading buffer (*NuPAGE 4X*, Invitrogen Life Technologies). For electrophoresis, digested samples, along with non-digested controls, were boiled at 100 °C for 10 min and loaded onto 4–12% acrylamide gels (*NuPAGE Midi gel*, Invitrogen Life Technologies). Electrophoresis was performed for 1 h 20 min (10 min at 70 V, 10 min at 110 V, and 1 h at 150 V). The gel was then transferred to a glass recipient and stained using Blue-Safe (NZYtech) for total protein staining for at least 1 h at room temperature and with gentle agitation. To test the effect of dextran sulfate during digestion, cofactor-free PMSA products were first incubated with 0.5% (w/v) dextran sulfate for 1 h at room temperature with gentle rotation, then digested with PK and visualized by total protein staining as with other samples.

## Preparation of prion-coated beads for long-term storage of PMSA products and their biochemical characterization

Prion-coated beads were prepared as described previously [47]. Zirconia-silicate beads of 1 mm and 2.3 mm diameter (BioSpec Products Inc.) were rinsed three times with PBS to eliminate impurities or dust from their surfaces that could influence prion adsorption efficiency. After a final rinse with distilled, Milli-Q grade water (Elix, Millipore), the beads were dried completely in a laboratory oven at 42 °C. For the coating process, approximately 1 ml of PMSA reaction product was placed into 2-ml tubes containing approximately 21 g of 1 mm beads (equivalent to a surface area of about 500 mm²) (2.1 g in case of the small tubes). The tubes were placed on a shaker and incubated for 1–2 h with gentle agitation (40 rpm) at room temperature. After the incubation step, the supernatant was removed, and the beads were subjected to a 24-h PMSA round using 800 μl of fresh substrate. The PMSA reactions were performed in 2-ml tubes as described in the corresponding section (*vide supra*) without addition of glass beads.

## Biochemical characterization

These assays were performed for the biochemical characterization of spontaneously misfolded recombinant mouse L108I PrP seeds generated in the presence of dextran sulfate (rec-MoL108I-Dx seeds, designated as stMI-01, stMI-03, btMI-05, and btMI-09). The seeds were subjected to proteinase K resistance assay, propagation capacity analysis on homologous and heterologous rec-PrP containing substrates, and transmission electron microscopy imaging.

*Protease K digestion assay:* An exclusive proteinase K (PK) (Roche) stock solution was prepared for this assay to avoid potential variability due to different PK preparations. 400 μl of each rec-MoL108I Dx seeds were digested at PK concentrations of 25, 100, 500, 1,000 and 2,000 μg/ml of PK for 45 min at 42 °C in a laboratory oven. After digestion, rec-PrP$^{res}$ detection was performed through electrophoresis and total protein staining as described in the corresponding section (*vide supra*). For quantitative assessment of relative PK resistance, three independent digestion reactions were performed with smaller sample volumes under identical conditions, following by Western blotting and densitometric analysis of rec-PrP$^{res}$ signal intensities. Densitometric analysis was conducted using ImageJ software, with values normalized to samples digested with 25 μg/ml of PK (set as 100%, signal intensity).

*Propagation capacity in homologous rec-PrP containing substrate by serial dilution:* rec-MoL108I-Dx seeds were serially diluted from 1:10$^1$ to 1:10$^{11}$ in PMSA substrate containing mouse L108I rec-PrP and dextran sulfate, supplemented with approximately 15 zirconia-silicate beads (1 mm diameter, 0.3 mg). All dilutions were subjected to a 24-hour PMSA round in quadruplicate as described above. Formation of rec-PrP$^{res}$ in each tube was evaluated through proteinase K digestion, electrophoresis, and total protein staining as described in the corresponding section (*vide supra*).

*Propagation capacity in a heterologous substrate:* In this case each rec-MoL108I-Dx seed was diluted at 1:10 in triplicate in PMSA substrate containing wild type recombinant mouse PrP (L108), dextran sulfate, and 0.3 mg of 1 mm diameter zirconia-silicate beads. All the samples were then subjected to a 24-h PMSA round, and resulting products were analyzed by PK digestion, electrophoresis, and total protein staining as described in the corresponding section (*vide supra*).

*Negative-stain Transmission Electron Microscopy:* For the characterization by transmission electron microscopy and negative staining of the four rec-MoL108I Dx seeds, they were first propagated in PMSA using substrates complemented with 0.02% (w/v) of sodium dodecyl sulfate (SDS, Sigma-Aldrich). After a 4-h PMSA amplification round, seeded with

2 zirconia silicate beads of 2.3 mm loaded with rec-PrP$^{res}$ of each preparation as explained above, a second serial 20-h PMSA round was performed using all the product generated in the previous 4 h-round as seed, diluted 1:5 in fresh substrate, again complemented with 0.5% dextran sulfate, 0.02% SDS and up to 15 unloaded zirconia silicate beads of 2.3 mm diameter to favor propagation. The resulting samples were then processed as follows: SDS concentration was increased up to 0.1% (w/v) and samples incubated for 3 h at room temperature (RT) with gentle rotation. Then, they were centrifuged at 100 $g$ for 30 min at RT to remove any debris that may have been released from the beads during the agitation process. The supernatant was recovered and subjected to a second centrifugation at 1,000 $g$ for 30 minutes at RT, after which it was discarded. The resulting pellet was then resuspended in PBS with 0.1% SDS and incubated overnight at RT on a rotary shaker at low speed. The samples were then digested with proteinase K at 50 µg/ml for 1 h at 42 °C without shaking and centrifuged immediately after at 1,000 $g$ for 30 min at RT. The supernatant was discarded, and each pellet resuspended thoroughly in a 10 mM Tris-HCl (Fisher Bioreagents) and 10 mM NaCl (Sigma-Aldrich) solution (Tris-NaCl) containing 0.05% (w/v) of SDS for washing. Samples were then centrifuged for 10 min at 600 $g$ and RT, and the pellet was washed again with a 10 mM Tris-NaCl solution containing 0.02% of SDS. After a final centrifugation of all four samples for 10 min at 600 $g$ and RT, pellets were resuspended in Tris-NaCl containing 0.02% of SDS, achieving 500-fold concentrated samples. For imaging, the samples were deposited onto copper grids (CF400-CU, Electron Microscopy Sciences) with a thin carbon film previously subjected to ionic discharge to hydrophilize the surface. The grid was positioned so that the carbon film was in contact with the sample and incubated in this position for 1 min at room temperature. Subsequently, the grid was placed on milli-Q grade water to wash off the excess and the liquid was removed by touching the edge of the grid with filter paper. Finally, the grids were deposited onto drops of 2% (w/v) uranyl acetate, incubated for 45 s for staining. The grids were dried at room temperature on filter paper before visualization. Images were acquired in a JEM-1230 (JEOL) transmission electron microscope stabilized at 100 kV and equipped with an Orius SC1000 CCD camera (GATAN).

### In vitro propagation of PMSA products by PMCA

*Preparation of PMCA substrates*: Brain PrP$^C$-based substrates from transgenic mice expressing approximately 3-fold the murine L108I PrP (TgMoL108I 3x), were prepared as previously described [39]. Briefly, perfused whole brains were homogenized at 10% (w/v) in Conversion Buffer (CB) with protease inhibitor cocktail (Roche) using a glass potter pestle (Fisher scientific), aliquoted, and stored at −80 °C until required.

*Brain-PMCA*: This assay was based on slightly modified versions of the PMCA previously described [28,48] and was performed to estimate the potential infectivity of the recombinant seeds generated by PMSA prior to bioassays [39]. For that, the four rec-MoL108I Dx seeds were first partially purified by ultracentrifugation in density gradients. Briefly, the rec-PrP$^{res}$ from 15 ml of each PMSA product was concentrated by centrifugation at 19,000 $g$ and 4 °C for 15 min. 2 ml of the concentrated rec-PrP$^{res}$ were then loaded on top of continuous sulfate cesium (Sigma-Aldrich) gradients ranging from 1 M to 1.7 M, prepared in PBS using a gradient mixer (Sigma-Aldrich), all placed in Thinwall Ultra-Clear 13.2 ml centrifuge tubes (Beckman Coulter). After ultracentrifugation at 210,000 $g$ for 15 h at 20 °C using a SW41 Ti Swinging bucket rotor (Beckman Coulter) and an Optima L-90K ultracentrifuge (Beckman Coulter), a visible protein halo could be detected in all cases. The fraction(s) presenting a visible halo were transferred to clean 5 ml Eppendorf tubes and diluted up to 5 ml with MilliQ-grade water for washing. They were centrifuged at 4,000 $g$ for 30 min and the

rec-PrP$^{res}$-containing pellets were resuspended in 1 ml of PBS, transferred into 1.5 ml Eppendorf tubes and submitted to centrifugation again at 19,000 $g$ for 15 min, repeating the PBS resuspension and centrifugation steps twice. Finally, the purified fractions were resuspended in 25–75 µl of PBS and used as seed in PMCA reactions. The four seeds were diluted at 1:10 in the substrate prepared with TgMoL108I (3x) brain homogenates for the first 24 h round of PMCA, performed in a S-4000 Misonix sonicator (Qsonica) with incubation cycles of 30 min followed by sonication pulses of 20 s at 80% power and keeping the temperature at 38 °C through a circulating water bath. To avoid cross-contamination, all PMCA tubes were sealed with plastic film (Parafilm). A total of three serial rounds of PMCA were carried out for each seed, diluting the product of the previous round 1:10 in fresh substrate for an additional 24-h PMCA round performed under identical conditions. After each round of PMCA, the external surfaces of all the tubes were cleaned thoroughly with sodium hypochlorite and tubes containing different seeds or from different experiments were treated separately. 1 mm zirconium silicate beads (BioSpec Products) were included in each reaction to favor propagation as well as unseeded controls, which were subjected also to 3 serial PMCA rounds together with all the seeded samples [45]. Additionally, a tube seeded with each of the PMSA products at 1:10 dilution but not subjected to PMCA was included as control of the basal signal of each seed at first passage. Prion propagation was determined by PK-digestion of the PMCA products followed by Western blotting as explained below.

## PrP$^{Sc}$ detection by electrophoresis and Western blot

*Proteinase K digestion of brain homogenate samples:* For PrP$^{Sc}$ detection in PMCA products or brain homogenates from inoculated animals, PK (Roche) digestion was performed prior to Western blotting. PMCA products or previously homogenized brains of inoculated animals [at 10% (w/v) in PBS (Fisher Bioreagents) with Protease inhibitor cocktail (Roche)] were mixed with digestion buffer [2% (w/V Tween-20 (Sigma-Aldrich), 2% (v/v) NP-40 (Sigma-Aldrich) and 5% (w/v) Sarkosyl (Sigma-Aldrich) in PBS] at 1:1 (v/v). PK was added to reach a final concentration of 170 µg/ml in each sample, which were incubated at 42 °C for 1 h with shaking at 450 rpm in a thermomixer (Eppendorf). Digestion was stopped by adding loading buffer (NuPage 4x Loading Buffer, Invitrogen Life Technologies) 1:3 (v/v) and boiling samples for 10 min at 100 °C.

*Western blot:* Digested samples, together with non-digested controls, were boiled at 100 °C for 10 min and loaded onto 4–12% acrylamide gels (*NuPAGE Midi gel*, Invitrogen Life Technologies) for 1 h 20 min (10 min at 70 V, 10 min at 110 V and 1 h at 150 V). Samples were then transferred to PVDF membranes using the iBlot 3 (Invitrogen Life Technologies). The membranes were developed using the iBind Flex Western Device (Invitrogen Life Technologies) and Sha31 (1:4,000) monoclonal antibody (Cayman chemical). After incubation with peroxidase-conjugated secondary anti-mouse antibody (m-IgGκ BP-HRP, Santa Cruz Biotechnology), membranes were developed with an enhanced chemiluminescent horseradish peroxidase substrate (West Pico Plus, Thermo Scientific). An iBright CL750 imaging system (Invitrogen Life Technologies) was used for image acquisition and the software AlphaView (Alpha Innotech) for image processing.

## Bioassays/in vivo infectivity

*Preparation of inocula:* rec-PrP$^{res}$ products spontaneously generated by PMSA were diluted 1:10 in sterile PBS prior to intracerebral inoculation into TgMoL108I (3x), TgVole (1x), and wild-type mice (C57BL/6). Similarly, 10% brain homogenates from mice inoculated with classical prion strains such as RML [49] and 22L (TSE Resource Centre, Compton, Newbury, UK) were used as positive controls. The inocula for the second passage were prepared in all cases

as 1% (w/v) brain homogenates in PBS from the first passage animals in each model, selecting the brain homogenate of an animal from each group showing an incubation period as close as possible to the group mean.

*Animal inoculations*: Groups of 5 to 6-week-old TgMoL108I (3x), TgVole (1x) and wild-type mice (C57BL/6) were inoculated intracerebrally with 20 µl of recombinant inocula into the left cerebral hemisphere using a sterile disposable 27-gauge hypodermic needle (Terumo). Mice were anesthetized with injectable ketamine/medetodomidine anesthesia (75/1 mg/kg) (Imalgene 1000, Boehringer Ingelheim/ Domtor, Ecuphar), which was subsequently reverted using atipamezole hydrochloride (1 mg/kg) (Antisedan, Ecuphar). Alternatively, gaseous anesthesia (Isoflurane, IsoVet, Braun) was used when possible. Experimental groups were comprised of 4 to 9 animals each as indicated in Results section tables. The animals were examined twice a week until the development of neurological clinical signs, after which they were examined daily. Clinically affected animals were culled at an advanced stage of disease, but before neurological impairment compromised their welfare, by exposure to a rising concentration of carbon dioxide. The clinical signs monitored included kyphosis, gait abnormalities, altered coat state, depressed mental state, flattened back, eye discharge, hyperactivity, loss of body condition, and incontinence. Animals exhibiting two or more severe signs or invalidating motor disturbances were euthanized. Survival time was calculated as the interval between inoculation and culling or death. The brain was extracted immediately after culling and divided longitudinally; one half of the brain was kept frozen at −80 °C for biochemical analysis and the other part was placed into 10% phosphate-buffered formalin solution (Sigma-Aldrich) for histopathological analysis.

## Statistical analysis

Statistical analyses were performed using GraphPad Prism version 9.0 (GraphPad Software, San Diego, CA, USA). Incubation period data are presented as mean ± standard error of the mean (SEM). Differences in relative proteinase K resistance were analyzed using a Saphiro-Wilk test to assess distribution, followed by Friedman test with Wilcoxon signed-rank tests for post hoc comparisons. Differences in survival times between groups were analyzed using Kaplan-Meier survival curves, and Saphiro-Wilk test was employed to determine data distribution. For experiments in which all groups show normal distribution, multiple group comparisons of post-inoculation times were performed using one-way ANOVA followed by Tukey's post-hoc test. In the cases in which data showed non-normal distribution, statistical significance was determined using Mann-Whitney U test. Statistical significance was set at $p < 0.05$, in all cases. Sample sizes were determined based on previous similar studies to ensure adequate statistical power while adhering to the principles of reduction in animal experimentation.

## Anatomopathological analysis and immunohistochemistry

Transversal sections of the formalin-fixed half brain were performed at the levels of the medulla oblongata, piriform cortex, and optic chiasm. All sections were then dehydrated dehydration through increasing alcohol concentrations and xylene before being embedded in paraffin-wax. of Four-micrometers sections were placed on glass microscope slides and stained with hematoxylin (Sigma-Aldrich) and eosin (Casa Álvarez). Additional sections were mounted on 3-trietoxysilil-propilamine-coated glass microscope slides (DAKO) for immunohistochemistry, as previously described [50]. Treated tissue sections were deparaffinized and subjected to epitope unmasking treatments: immersion in formic acid and boiling (at pH 6.15) in a pressure cooker, followed by pre-treatment with 4 µg/ml of proteinase K (Roche).

Endogenous peroxidases were blocked by immersion in a 3% $H_2O_2$ in methanol solution. Sections were then incubated overnight with anti-PrP monoclonal antibody 6C2 (1:1000, CVI-Wageningen UR) or, exceptionally, with 2G11 (1:100, Bertin Pharma), and visualized using the Goat anti-mouse EnVision system (DAKO) and 3,3'-diaminobenzidine (Sigma Aldrich). As a background control, incubation with the primary antibody was omitted. Brain histological lesions, spongiform change, and PrP$^{res}$ immunolabeling were evaluated under a light microscope. Using a semi-quantitative approach [51], spongiform lesion and PrP$^{res}$ immunolabeling were separately scored in fourteen different brain regions: piriform cortex (Pfc), hippocampus (H), occipital cortex (Oc), temporal cortex (Tc), parietal cortex (Pc), frontal cortex (Fc), striatum (S), thalamus (T), hypothalamus (HT), mesencephalon (M), medulla oblongata (Mobl), cerebellar nuclei (Cm), cerebellar vermis (Cv), and cerebellar cortex (Cc). Scores ranging from (0) absence of spongiosis or immunolabeling to (4) maximum intensity of lesion or immunolabeling were assigned to each brain area studied, which was investigated globally as a region for scoring. The scores were: (0) absence, (1) mild, (2) moderate, (3) intense, and (4) maximum intensity of lesion or immunolabeling. Brain profiles were plotted as a function of the anatomical areas, which were ordered representing the caudo-rostral axis of the encephalon. Graphs were plotted using Microsoft Office Excel software.

## Results

### Generation of recombinant PrP variants with all naturally occurring amino acids at position 108

Previous work with recombinant bank vole PrP (rec-PrP) bearing an isoleucine polymorphism at position 109 led to the development of a method allowing consistent spontaneous misfolding of this protein *in vitro* into *bona fide* prions [45]. Given the high susceptibility of bank voles to many prion diseases [52] and the development of a sporadic prion disease in transgenic mice overexpressing this protein bearing an isoleucine polymorphism at position 109 [18], we investigated whether the presence of isoleucine at this position is critical for such behavior and if spontaneous misfolding could occur using recombinant PrP from other species. For that, we used recombinant mouse PrP and evaluated the relevance of the amino acid at position 108 (equivalent to position 109 in bank vole PrP). We generated 20 recombinant mouse PrP variants, each with a different naturally occurring amino acid residue at the target position (primers used for mutagenesis listed in S1 Table). After expression and purification, we prepared PMSA substrates for each rec-PrP variant. The rec-PrP concentration in each substrate was adjusted using BCA assay and verified through electrophoresis and total protein staining (S1 Fig).

### Mouse L108I PrP shows an enhanced capacity for spontaneous misfolding

We subjected the 20 substrates to PMSA, as developed for spontaneous misfolding of bank vole rec-PrP, to evaluate the misfolding proneness of each variant. Using 4 replicate tubes for each substrate, all complemented with 1 mm diameter glass beads and containing dextran sulfate as cofactor, we assessed the percentage of replicates in which proteinase K (PK)-resistant misfolded rec-PrP (rec-PrP$^{res}$) could be detected after 24 h of PMSA by PK digestion, electrophoresis, and total protein staining of all PMSA products.

As summarized in Fig 1A (with representative examples shown in Fig 1B), four mouse rec-PrP variants were able to misfold spontaneously in 24 h, including, as expected, mouse L108I rec-PrP. Apart from isoleucine, the presence of histidine, lysine, and methionine also permitted spontaneous misfolding of mouse rec-PrP, whereas all the rest, including the wild-type, mouse L108, could not be misfolded under the same conditions. The rec-PrP$^{res}$

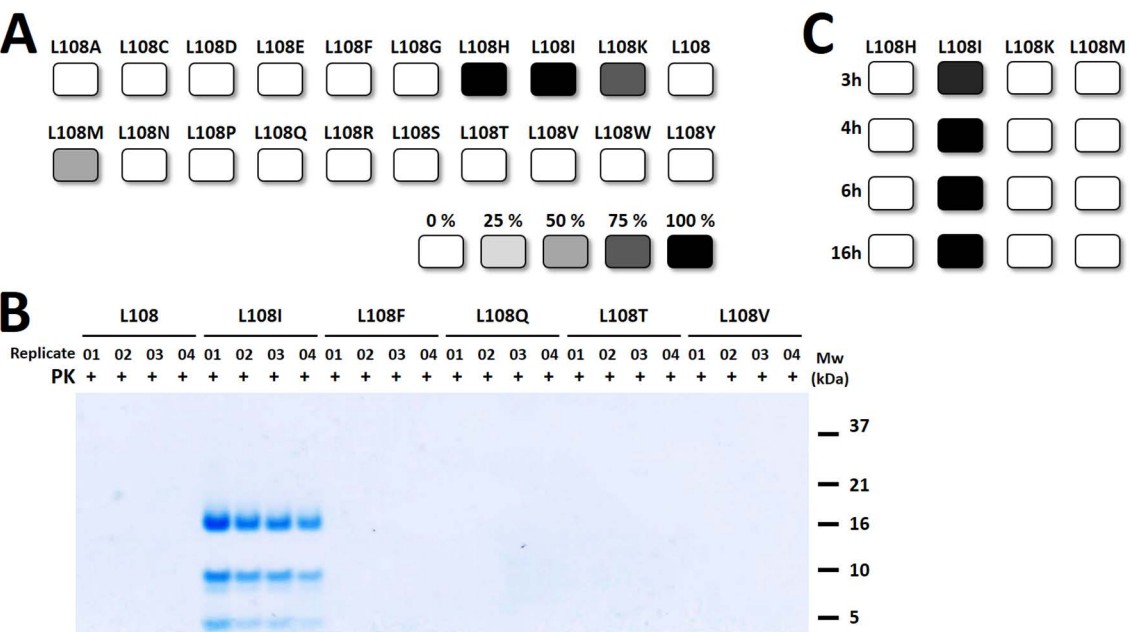

**Fig 1. Evaluation of spontaneous misfolding capacity of 20 mouse rec-PrP variants with naturally occurring amino acid substitutions at position 108. A)** PMSA experiments were conducted on 20 substrates, each containing a distinct rec-PrP variant. After a 24-hour PMSA round (four replicates per substrate), rec-PrP$^{res}$ formation was assessed via PK digestion, electrophoresis, and total protein staining. Results, displayed in grayscale, show four variants capable of spontaneous misfolding: L108I and L108H (100% positive replicates), L108K (75%), and L108M (50%). Variants without misfolding propensity showed no detectable rec-PrP$^{res}$ (white). **B)** An electrophoresis gel illustrates rec-PrP$^{res}$ detection for selected variants (L108 wt, L108I, L108F, L108Q, L108T, and L108V). Only L108I formed rec-PrP$^{res}$ with the characteristic 16 kDa PK-resistant band expected of recombinant prions. **C)** Further PMSA experiments with the four misfolding-prone variants (L108I, L108H, L108K, and L108M) were conducted with shorter reaction times (3, 4, 6, and 16 hours). Results indicate that only L108I misfolded under these restrictive conditions, confirming it as the most prone to spontaneous misfolding among all variants. MW: Molecular weight marker.

resulting from the proteins bearing histidine, lysine, and methionine showed a highly coincidental electrophoretic mobility pattern with the isoleucine-bearing PMSA products. All displayed a predominant fragment of approximately 16 kDa and two intermediate fragments of around 10 and 8 kDa, with the latter appearing fainter in L108H and L108I compared to the other two proteins. Additional small molecular weight fragments below 5 kDa were observed, showing greater variability among different preparations. Among the four variants that could be misfolded, some differences were detected. Mouse L108I and L108H rec-PrP, with all replicates positive after 24 h of PMSA, showed a higher misfolding efficiency than variants L108K and L108M, with 75 and 50% of positive replicate tubes, respectively. To further discriminate whether any of the four variants presented a greater proneness to spontaneous misfolding, we performed PMSA reactions of 3, 4, 6, and 16 h. Using this temporal restriction, mouse L108I variant was the only one giving rise to rec-PrP$^{res}$ as early as 3 h of reaction (Fig 1C), demonstrating a higher spontaneous misfolding capacity than the rest of the variants tested.

## Misfolding of mouse L108I rec-PrP in PMSA results in four potentially distinct conformers

As with bank vole rec-PrP, where distinct strains were formed spontaneously under identical conditions, the misfolding of mouse L108I rec-PrP also resulted in varied electrophoretic

patterns. Although the diversity of patterns was lower than that detected for bank vole PrP, we considered that there could be distinct strains with similar electrophoretic patterns. Therefore, four rec-PrPres containing PMSA products from independent experiments were randomly chosen for further characterization of differential strain properties. These were designated as stMI-01, stMI-03, btMI-05, and btMI-09 (Fig 2), where 'st' refers to 'small tubes,' and 'bt' to 'big tubes' (1.5 ml Eppendorf type tubes), indicating the type of tubes in which they were generated, while 'MI' stands for 'Mouse Isoleucine.'

## Biochemical characterization of the selected mouse L108I rec-PrPres indicates distinctive strain properties

Relative resistance to PK digestion was assessed using increasing enzyme concentrations from 25 µg/ml to 2000 µg/ml for 45 min at 42 °C. As shown in S2A Fig, both total protein staining and the Western blot of three independent replicates demonstrated high resistance in all four preparations. stMI-01, btMI-05, and btMI-09, showed similar resistance patterns with no statistically significant differences among them. However, stMI-03 exhibited significantly lower relative resistance compared to the other preparations (stMI-01 vs. stMI-03: p-value = 0.013; btMI-05 vs. stMI-03, p-value = 0.013; and btMI-09 vs. stMI-03, p-value = 0.028). All four products demonstrated high propagation capacity in a homologous substrate *in vitro*, with detectable rec-PrPres after serial dilutions up to $10^{-8}$ (S2B Fig). Slight differences in propagation efficiencies were observed but may not necessarily indicate distinct strain due to the high dependence on initial prion titers. To evaluate differential host range, we tested the propagation capacity on a heterologous substrate (wild-type recombinant mouse PrP) using prion-coated zirconia-silicate beads as seeds. Only stMI-03 was unable to propagate in this substrate, indicating differential biochemical properties and likely representing a distinct strain (S2C Fig).

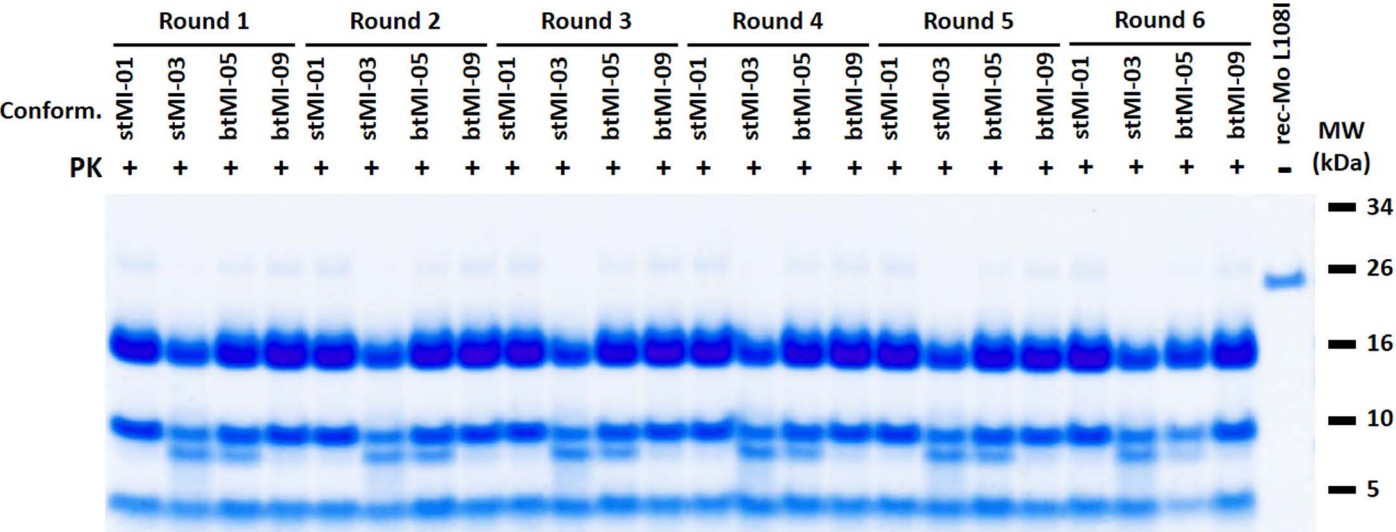

**Fig 2. Generation and stable propagation of four potentially distinct conformers of spontaneously misfolded mouse L108I rec-PrPres.** Four spontaneously generated PMSA products (stMI-01, stMI-03, btMI-05, btMI-09) were selected as potentially distinct conformers. These were produced after a single 24-hour PMSA round in the presence of 0.5% dextran sulfate and assessed by PK digestion, electrophoresis, and total protein staining. All four conformers could be stably propagated through serial PMSA rounds (1:10 seed:substrate dilutions), maintaining their characteristic electrophoretic profiles over six rounds. The undigested substrate (rec-MoL108I) is included for size comparison. Despite similar fragmentation patterns, further analyses reveal distinct properties among the conformers. Conform: Conformer; MW: Molecular weight marker.

## Characterization of selected mouse L108I rec-PrP$^{res}$ by electron microscopy reveals expected prion rods and heterogeneous arrangements

Transmission electron microscopy (TEM) with negative staining was performed to compare the ultrastructural arrangement of PMSA products. Representative micrographs (S3 Fig) show that all four preparations exhibit fibrillary arrangement reminiscent of prion rods observed in brain-derived preparations [53]. In all cases, fibers of varying lengths were observed with a certain tendency towards lateral association, hindering the assessment of fiber width, although many micrographs seem to contain approximately 12 nm wide fibers as majoritarian elements. While the four preparations appear similar due to the limited resolution of negative stain TEM, a detailed analysis of hundreds of micrographs revealed subtle differences. stMI-01 is characterized by at least two apparent types of fibers: straight and curved. The sample contains twisted fibers with potentially very long half-pitches, as well as apparently flat fibers, some without detectable torsion for over 400 nm of fiber length (see illustrative details in S3E Fig). Whether this flat and twisted fibers are constituted by the same or distinctive PrP$^{Sc}$ cores remains to be addressed. In contrast stMI-03 appears enriched in straight fibers, with the mixture of straight and curved fibers being less evident. btMI-05 and btMI-09 generally contain longer fibers compared to stMI-01 and stMI-03. btMI-09 is distinguished by the presence of a few very thin fibers not found in other PMSA products and a high number of very short rods. Despite these subtle differences, all samples predominantly contain apparently flat fibers or fibers with exceptionally long half-pitches, mixed with fibers displaying clear torsions and half-pitches of approximately 200 nm (S3E Fig). Many rods exhibit two axial densities of approximately 12 nm, resembling rail tracks. This could represent paired closely associated fibers or a single fiber with a central groove. In conclusion, all four samples display abundant fibers comparable to brain-derived prion rods, with apparent mixtures of two or three distinct fibrillary arrangements. This heterogeneity could indicate the presence of multiple conformers or strains despite showing a single electrophoretic pattern after PK digestion.

## Recombinant PMSA products can induce misfolding of PrP$^{C}$ from brain in PMCA

The spontaneously generated misfolded rec-PrP preparations were also used as seeds in PMCA experiments to examine whether they could propagate in a brain homogenate substrate containing brain-derived PrP$^{C}$, thereby indicating their potential infectivity *in vivo*. Additionally, this experiment aimed to assess if the preparations had distinct propagation capacities, which might suggest different strain properties. For that, brain homogenates from a transgenic mouse line overexpressing 3-fold mouse PrP with an isoleucine substitution at position 108 (TgMoL108I)—a PrP sequence homologous to that of the recombinant seeds— were used as substrates. The four PMSA products were partially purified by ultracentrifugation through a density gradient and then inoculated into the TgMoL108I brain homogenates. These partially purified rec-PrP$^{res}$ samples were diluted in the TgMoL108I brain homogenate and were subjected to three 24-h serial PMCA rounds. For the subsequent rounds, the product from the previous round was diluted 1:10 in fresh substrate. As shown in S4 Fig, all four PMSA products were able to induce misfolding of the brain-derived PrP$^{C}$ during the first PMCA round, indicating probable infectivity *in vivo*. However, since all four seeds exhibited similar propagation efficiency, this method did not distinguish whether they represent different prion strains.

Altogether, our analyses clearly differentiated stMI-03 from the rest (differential PK susceptibility and propagation capacity in heterologous substrate). However, the distinction between stMI-01, btMI-05 and btMI-09 remains unclear. The biochemical and *in vitro*

propagation analyses performed were insufficient to determine conclusively whether they are the same strain or distinct ones. Further investigation is necessary to establish potential strain differences.

### Recombinant PMSA products induce a prion disease in homologous PrP$^C$ overexpressing transgenic mouse model, confirming their bona fide prion nature

Based on the propagation ability of all four PMSA products in TgMoL108I brain homogenate, which indicated potential infectivity *in vivo*, we inoculated intracerebrally these products into TgMoL108I transgenic mice expressing homologous PrP sequence. This approach aimed to confirm their *bona fide* prion nature and definitively assess their distinct strain characteristics. As controls for the susceptibility of this animal model to classical brain-derived prion strains, we also inoculated intracerebrally the well-known RML and 22L murine prion strains. To compare relative infectivity between recombinant preparations and brain-derived prions, misfolded PrP amounts were estimated following methods previously used for bank vole recombinant prions [45]. Assuming variable conversion yield in PMSA ranging from 20 to 100% and taking into account that the PMSA substrates were prepared with approximately 0.1 mg/ml of rec-PrP, each PMSA preparation would contain 0.02 to 0.1 mg/ml of misfolded rec-PrP, which after dilution in PBS and considering the volume injected in the animals would suppose around 0.04 to 0.2 μg of misfolded rec-PrP. To estimate the prion load in the RML and 22L preparations, we assume that the titer in a prion infected animal at terminal stage is similar to that determined for 263k-infected hamsters, that is of around 20 μg/g of brain tissue [45], and thus, we injected approximately 0.004 μg, that is, 10 to 50-fold less than for the recombinant preparations. As expected, all PMSA products induced transmissible spongiform encephalopathy (TSE) in the inoculated animals, albeit with no statistically significant differences as determined by one-way ANOVA (p-value = 0.226) (Table 1). stMI-03, btMI-05 and btMI-09 showed 100% attack rates, whereas stMI-01 demonstrated an incomplete attack rate. We found classical, three-banded pattern PrP$^{Sc}$ in all animals exhibiting clinical signs (Fig 3A), with at least two distinctive electrophoretic profiles. Animals inoculated with stMI-01 and btMI-05 inoculated animals displayed an unglycosylated band of approximately 21 kDa, similar to that of 22L and RML inoculated animals, except for the relative glycosylation ratio. In contrast, in the mice inoculated with stMI-03 and btMI-09, two distinct and disparate profiles were detected in different individual animals, which could reflect the observations in TEM micrographs. This further indicates the potential existence of conformer mixtures. Fig 3B presents Kaplan-Meier survival curves for each of the PMSA products, further illustrating different behaviors among the potentially distinct PMSA products.

Although not statistically significant, the four PMSA products showed different incubation periods, potentially suggesting distinct strain properties or variations in prion titer. As this represents a first passage of recombinant prions *in vivo*, where a transmission barrier posed by fully glycosylated and GPI-anchored PrP may exist, a second passage would typically be required to firmly establish strain differences among the four PMSA preparations. However, a second passage in the same overexpressing model was avoided due to the TgMoL108I mice's propensity for spontaneous misfolding at advanced ages (mean disease onset: 339 ± 9 days, Table 1). This precaution was taken to prevent potential confounding effects that the spontaneous strain generated in TgMoL108I model could interfere with the recombinant preparation-induced pathology in a second passage. Fortunately, the spontaneous disease developed by some TgMoL108I mice shows atypical features, including low molecular weight PrP$^{Sc}$ fragments and a lesion profile clearly distinguishable from those induced by classical prion strains. Moreover, this atypical disease is not transmissible to wild-type mice

**Table 1. Intracerebral inoculation of spontaneously generated PMSA products with distinct electrophoretic migration patterns in TgMoL108I mice.**

| Strain | Origin[a] | TgMoL108I mice | | |
|---|---|---|---|---|
| | | Attack rate | Days post-inoculation (± SEM) | PrP$^{Sc}$ classic pattern (WB) |
| stMI-01 | rec-MoL108I Dx | 4/8[ƒ] | 260 ± 36[$] | 4/8[¤] |
| stMI-03 | rec-MoL108I Dx | 6/6 | 207 ± 11 | 6/6 |
| btMI-05 | rec-MoL108I Dx | 9/9 | 173 ± 4 | 9/9 |
| btMI-09 | rec-MoL108I Dx | 5/5 | 158 ± 5 | 5/5 |
| | | | | |
| RML | Brain homogenate | 5/5 | 85 ± 5 | 5/5 |
| 22L | Brain homogenate | 8/8 | 95 ± 1 | 8/8 |
| No inoculum | – | 18/18 | 339 ± 9[&] | 0/18 |

[a] **Brain homogenate:** Indicates that the seed used as inoculum originates from a 10% brain homogenate in PBS (phosphate-buffered saline) derived from an animal infected with the specified prion strain. The homogenates were further diluted 1:10 in PBS and 20 μl were inoculated intracerebrally in each animal. **rec-MoL108I Dx:** Indicates that the seed used as inoculum originates from the recombinant L108I mouse PrP protein, which was misfolded using the cofactor sulfated dextran, resulting in the indicated prion strain. The PMSA products were diluted 1:10 in PBS and 20 μl of these diluted preparations were inoculated intracerebrally in each TgMoL108I mouse.

[ƒ] The attack rate was calculated considering only the animals that were euthanized and exhibited a classic PrP$^{Sc}$ pattern on Western blot (¤) and or by IHC.

[$] The standard error of the mean (SEM) for this group of animals was calculated by including all euthanized animals, encompassing even those negative by Western blot that exhibited signs of developing a spontaneous disease.

[¤] TgMoL108I mice spontaneously develop a TSE at 339 ± 9 days, marked by the presence of atypical PrP$^{res}$ detectable by Western blot and characterized by a 7–10 kDa low molecular weight fragment. Consequently, the detection of PrP$^{Sc}$ with a classical pattern confirms that the observed signs are not attributable to the spontaneous disease associated with this model.

[&] This incubation time and its SEM refers to the onset of a spontaneous disease occurring in 100% of these animals, characterized by distinctive clinical signs and the presence of PrP$^{res}$ with an atypical pattern, detectable by both Western blot and IHC.

(see Table 2 and S5 Fig for further information). These characteristics allowed us to differentiate between spontaneous and induced pathologies. Given these factors, we decided to perform a second passage in wild-type mice to definitively assess potential strain differences, thereby avoiding complications associated with the TgMoL108I model while still addressing the need for strain characterization.

## Secondary transmission of TgMoL108I-passaged recombinant prions in wild-type mice reveals differential strain characteristics

Secondary transmission of TgMoL108I-passaged recombinant prions to wild-type (C57BL/6) animals revealed further differences among the four inocula, showing statistically significant differences between all the preparations except for stMI-01 and btMI-05 (Mann-Whitney U test, p-value = 0.635) (Table 2). All four showed 100% attack rates, with an expected prolongation of incubation periods in most cases, with the notable exception of stMI-03. This prolongation may be due to the transmission barrier between the two mouse PrP variants (wild-type and L108I), or the lower PrP$^{C}$ expression level in wild-type mice compared to the TgMoL108I model. Nonetheless, the TgMoL108I-passaged control inocula (RML and 22L) demonstrated similar incubation periods in wild-type mice compared to their direct transmission indicates a very weak effect of this polymorphic barrier. This similarity suggests that the effect of the isoleucine substitution at position 108 was minimal for brain-derived inocula, indicating that the

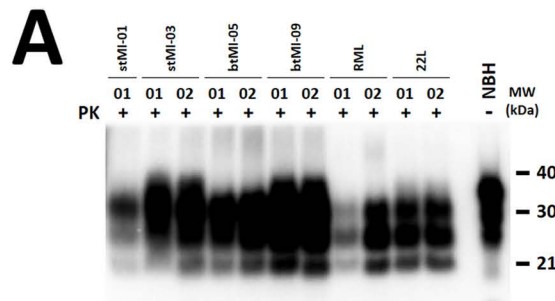

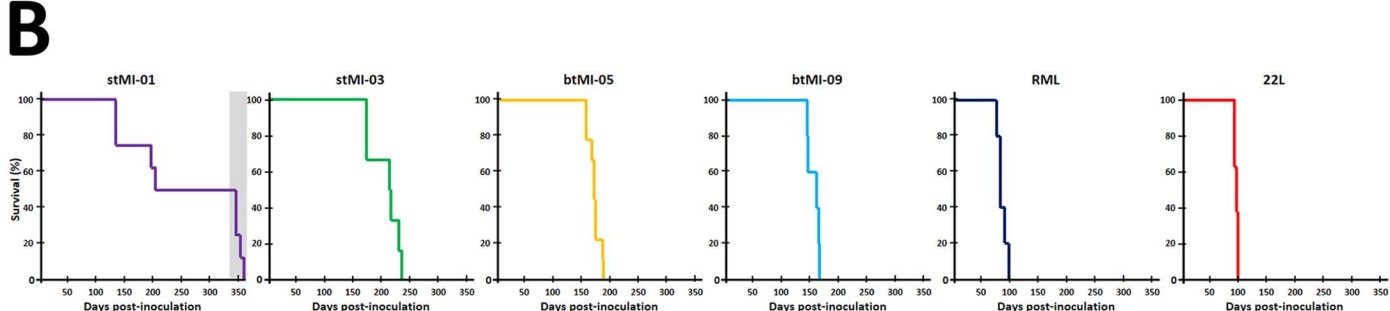

**Fig 3. A) Biochemical and survival analysis of TgMoL108I mice inoculated with spontaneously misfolded PMSA products.** Proteinase K digestion, electrophoresis, and Western blot (Sha31, 1:4,000) of brain homogenates from TgMoL108I mice inoculated with the four PMSA products (stMI-01, stMI-03, btMI-05, btMI-09, using brain-derived RML and 22L prions as controls) revealed the classical three-banded PrP$^{Sc}$ pattern, confirming their infectious nature as bona fide recombinant prions. Representative samples from each group and controls (brain-derived RML and 22L prions) were mostly indistinguishable, though slight electrophoretic differences were observed in stMI-03 (sample 01, higher bands) and btMI-09 (sample 02, lower bands), suggesting potential conformer mixtures. **B) Kaplan-Meier survival curves for animals inoculated with PMSA products and RML and 22L strains demonstrated variability in incubation periods.** stMI-01 and stMI-03 showed greater dispersion, while btMI-05 and btMI-09 exhibited more uniform profiles similar to RML/22L but with slightly longer incubation periods. These longer periods may reflect the need for adaptation or a transmission barrier between recombinant prions and brain-derived PrP$^C$, while higher dispersion could indicate strain-specific barriers, conformer mixtures undergoing selection, or immature prion conformers. Notably, the longest incubation for stMI-01 (~340 dpi) coincided with spontaneous atypical prion disease in the TgMoL108I model, distinct from induced TSE due to the absence of classical PrP$^{Sc}$ patterns. PK: Proteinase K; NBH: Normal brain homogenate; MW: Molecular weight marker.

transmission barrier posed by this amino acid change may be weak or nonexistent. Therefore, the prolongation of incubation periods observed for stMI-01, btMI-05 and btMI-09 responds to other reasons such as the pathobiological properties of the recombinant prions, that could require in this cases adaptation or selection of the TgMoL108I-passaged isolates. However, the secondary transmission of these control inocula into wild-type mice demonstrates that, apart from variations in incubation periods due to the overexpression in TgMoL108I mice, their fundamental strain properties remain unaltered. Intriguingly, stMI-03 inoculated animals exhibited a paradoxical phenomenon. These animals showed a shortening of the incubation period in wild-type mice compared to the primary inoculation in TgMoL108I, suggesting unique adaptive properties of this particular strain.

A detailed anatomopathological analysis of the brains from wild-type mice showing neurological signs upon inoculation with TgMoL108I-passaged recombinant products (first third of Table 2) was carried out to characterize potential strain differences among the four recombinant preparations. The evaluation of the brains of wild-type mice inoculated with TgMoL108I-passaged stMI-01 was incomplete due to suboptimal condition of the fixed samples. However, it was sufficient to conclude that it causes distinct lesions compared to other inocula, characterized by moderate spongiform lesions in the brainstem, mild in the cerebral cortices and practically absent in the cerebellum (with only one animal showing

**Table 2.** Intracerebral inoculation in C57BL/6 mice using brain products from TgMoL108I inoculated animals or spontaneously generated PMSA products.

| Strain | Passage | Origin[a] | C57BL/6 | | |
| --- | --- | --- | --- | --- | --- |
| | | | Attack rate | Days post-inoculation ($\pm$ SEM) | PrP$^{Sc}$ classic pattern (WB) |
| stMI-01> TgMoL108I | 2nd | **Brain homogenate** | 7/7 | 561 $\pm$ 26 | 7/7 |
| stMI-03> TgMoL108I | 2nd | **Brain homogenate** | 8/8 | 133 $\pm$ 6 | 8/8 |
| btMI-05> TgMoL108I | 2nd | **Brain homogenate** | 4/4 | 484 $\pm$ 85 | 4/4 |
| btMI-09> TgMoL108I | 2nd | **Brain homogenate** | 4/7[ƚ] | 191 $\pm$ 15[$] | 4/7 |
| RML> TgMoL108I | 2nd | **Brain homogenate** | 7/7 | 158 $\pm$ 1 | 7/7 |
| 22L> TgMoL108I | 2nd | **Brain homogenate** | 7/7 | 160 $\pm$ 6 | 7/7 |
| No inoculum> TgMoL108I[#] | 2nd | **Brain homogenate** | 0/6 | > 500 | 0/6 |
| No inoculum> TgMoL108I> WT | 3rd | **Brain homogenate** | 0/5 | > 500 | 0/5 |
| | | | | | |
| stMI-01 | 1st | **rec-MoL108I Dx** | 0/7 | > 500 | 0/7 |
| stMI-03 | 1st | **rec-MoL108I Dx** | 3/5[ƚ] | 472 $\pm$ 81 | 3/5 |
| btMI-05 | 1st | **rec-MoL108I Dx** | 6/6 | 197 $\pm$ 21[¤] | 6/6 |
| btMI-09 | 1st | **rec-MoL108I Dx** | 5/5 | 174 $\pm$ 15[X] | 5/5 |
| RML | 1st | **Brain homogenate** | 5/5 | 172 $\pm$ 4 | 5/5 |
| 22L | 1st | **Brain homogenate** | 5/5 | 175 $\pm$ 1 | 5/5 |
| | | | | | |
| stMI-01> WT | 2nd | **Brain homogenate** | ND | ND | ND |
| stMI-03> WT | 2nd | **Brain homogenate** | 7/7 | 130 $\pm$ 2 | 7/7 |
| btMI-05> WT | 2nd | **Brain homogenate** | 8/8 | 130 $\pm$ 3 | 8/8 |
| btMI-09> WT | 2nd | **Brain homogenate** | 6/6 | 128 $\pm$ 2 | 6/6 |

[a] **Brain homogenate:** Indicates that the seed used as inoculum originates from a 10% brain homogenate in PBS (phosphate-buffered saline) derived from an animal infected with the specified prion strain. The homogenates were further diluted 1:10 in PBS and 20 μl were inoculated intracerebrally in each animal. **rec-MoL108I Dx:** Indicates that the seed used as inoculum originates from the recombinant L108I mouse PrP protein, which was misfolded using the cofactor sulfated dextran, resulting in the indicated prion strain. The PMSA products were diluted 1:10 in PBS and 20 μl of these diluted preparations were inoculated intracerebrally in each TgMoL108I mouse.

[ƚ] The attack rate was calculated considering only the animals that were euthanized and exhibited a classic PrP$^{Sc}$ pattern on Western blot ([¤]) and/or by IHC.

[$] In this group, 3 animals exhibited clinical signs consistent with prion disease, but their PrP$^{res}$ analysis returned negative results in both Western blot and IHC tests. It is possible that these animals developed an intercurrent disease before the onset of genuine prion disease, which cannot be ruled out.

[#] Brain homogenates from non-inoculated TgMoL108I mice, showing signs of spontaneous disease were inoculated in C57BL/6 mice, that were culled after 500 days without signs of disease, and used for a subsequent passage in the same mice (row below), that did not show any sign of prion infection.

[¤] In this group, 3 animals developed intercurrent diseases and were excluded from the calculations of attack rate and incubation periods.

[X] In this group, 2 animals developed intercurrent diseases and were excluded from the calculations of attack rate and incubation periods.

intense spongiosis in the hypothalamus) and, in all animals, PrP$^{res}$ deposits were undetectable through IHC. Mice inoculated with TgMoL108I-passaged stMI-03 exhibited a homogeneous lesion profile with intense spongiosis in the thalamus and mesencephalon, and slightly milder in the medulla oblongata and hypothalamus. In this case, PrP$^{res}$ colocalized with spongiosis, being especially abundant in the thalamus and showing an intracytoplasmic granular pattern,

along with a diffuse immunostaining in the neuropil of the most affected areas. The analysis of btMI-05-inoculated animals also showed a homogeneous lesion profile in all animals except one. Most animals displayed moderate spongiform lesions in the brainstem and mild in the cerebral cortices and cerebellum, resembling those from stMI-03 inoculum. Among the animals inoculated with btMI-09, no spongiform lesion or PrP^res deposit could be detected in 3 out of 7 mice, despite obtaining positive results by Western blotting. However, the other 4 animals, on which the lesion profile is based, showed a similar profile to the stMI-03 and btMI-05-inoculated animals. The PrP^res in this case also colocalized with the spongiform lesions, characterized by synaptic fine punctate deposits, milder in neocortex ([Fig 4]). Thus, although stMI-01 is clearly distinct to the rest, these results do not allow further distinction between stMI-03, btMI-05, and btMI-09. Additionally, these three showed high intragroup variability possibly responding to the mixture of conformers previously hinted by electron microscopy and reflected in the subtle distinctive characteristics of some PrP^res detected by Western blot in certain TgMoL108I-passaged samples ([Fig 3]).

## Direct inoculation of recombinant prions in C57BL/6 demonstrates their ability to induce disease in wild-type animals with exceptional speed

We next evaluated if the spontaneously generated recombinant prions could directly infect wild-type mice, thereby providing unequivocal proof of their bona fide nature while avoiding potential artifacts from the TgMoL108I model.

We performed intracerebral inoculations of stMI-01, stMI-03, btMI-05, and btMI-09 recombinant prions in C57BL/6 mice. As shown in the central part of [Table 2], all inocula except stMI-01 caused a prion disease in inoculated animals, with stMI-03 resulting in incomplete attack rates, suggesting a greater transmission barrier for these strains, in contrast to the PMSA results where stMI-03 was unable to propagate. Statistical analysis of their incubation periods revealed significant differences between stMI-03 and the rest (one-way ANOVA, p-value = 0.0084), while no significant difference was observed for btMI-05 and btMI-09 (post-hoc Tukey's test, p-value = 0.916). Biochemical analysis of diseased animal brains revealed classical PrP^Sc electrophoretic profiles ([Fig 5A]). These patterns showed no detectable differences among them but were clearly different from the well-known RML and 22L strains, with a higher molecular weight monoglycosylated band and predominance of the diglycosylated one. Kaplan-Meier survival curves illustrated the stabilization of recombinant prion-induced strains btMI-05 and btMI-09, given the low dispersion in incubation periods ([Fig 5B]).

A second passage in wild-type mice was performed for stMI-03, btMI-05, and btMI-09 to stabilize the prion strains and complete strain typing. This secondary transmission (bottom third of [Table 2]) resulted in a notable reduction of incubation periods, likely due to overcoming transmission barriers posed by the amino acid sequence mismatch and differences between recombinant and brain-derived PrP, although without a third passage it is not possible to confirm full adaptation and stability of the recombinant prion preparations. All three strains showed incubation periods of around 130 days post inoculation, faster than any known mouse-adapted prion strain, with no statistically significant differences according to the Mann-Whitney U test results. Detailed histopathological analysis of second-passage brains confirmed all characteristics expected for prion disease, further validating the *bona fide* nature of recombinant prions ([S6 Fig]). Lesion profiling revealed similar lesions to those observed in wild-type mice inoculated with TgMoL108I-passaged inocula. Spongiform lesions were mild to moderate but constant in the neocortex, with marked lesions in the thalamus, midbrain, and in cerebellar cortex (this last feature, the cerebellar involvement, was also observed for strains stMI-03, btMI-05 and btMI-09 in wild-type mice after TgMoL108I passage, but not in stMI-01). PrP^res IHC staining was very faint in all three groups of animals, especially in

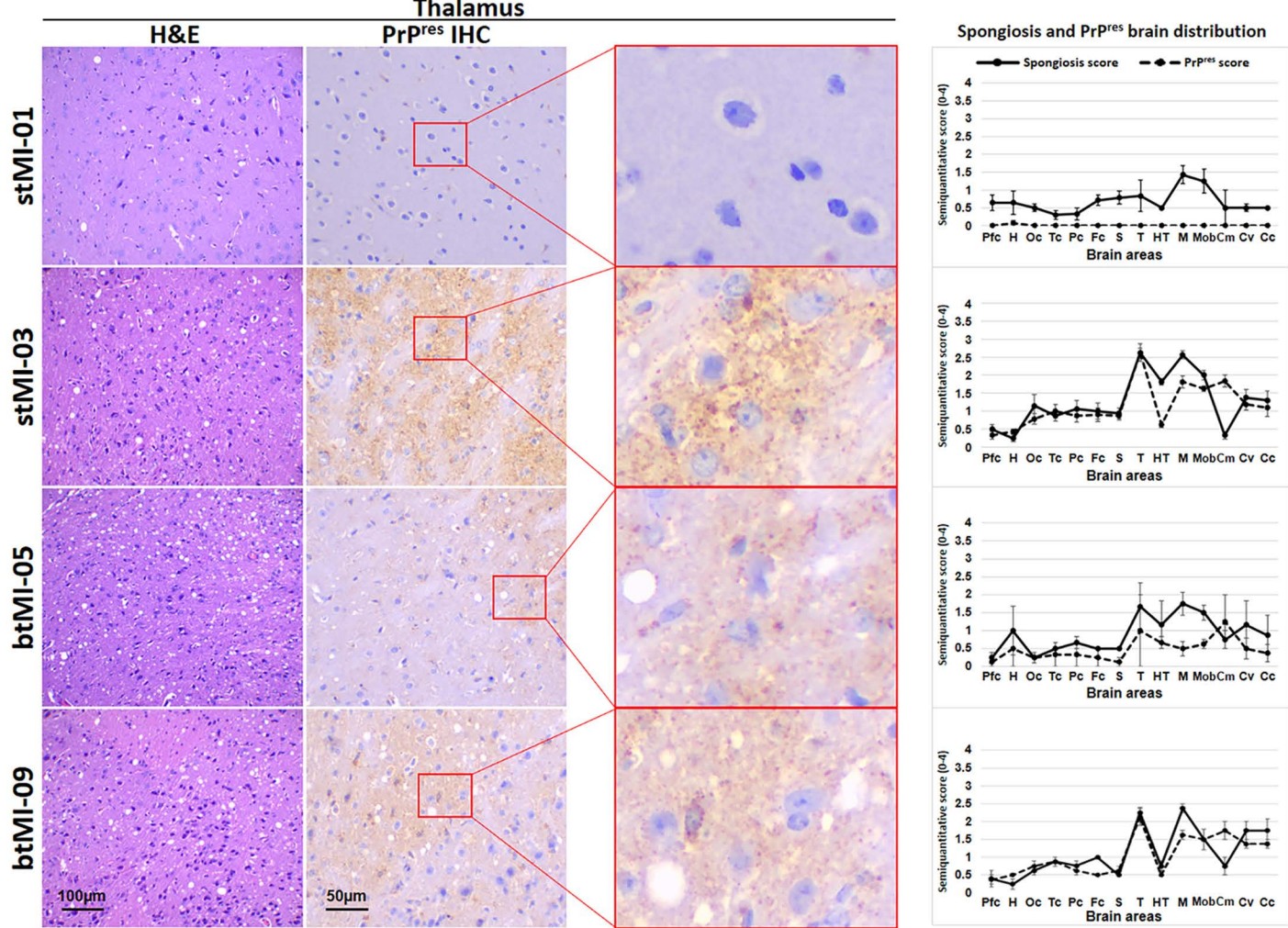

**Fig 4. Brain lesion and PrP^res deposit distribution of the TgMoL108I-passaged recombinant PMSA products after secondary transmission in wild-type mice.** Histopathological assessment of spongiform lesions and PrP^res deposits shows mild to moderate spongiform changes upon hematoxylin and eosin staining (H&E), as seen in the thalamic region of the brains of representative animals from each group. PrP^res deposits were labeled with 6C2 mAb (1:1,000) for stMI-01 and btMI-05, and 2G11 mAb (1:100) for stMI-03 and btMI-09. Deposits were detectable only for stMI-03, btMI-05, and some of the btMI-09-inoculated animals, all showing synaptic fine punctate deposits (see digitally enlarged images on the left). Spongiform lesion profiles and PrP^res deposition profiles, shown on the right, represent the mean semi-quantitative scoring (0–4, vertical axis, ± standard error of the mean -error bars-) of the spongiform lesions (continuous line, black) and the immuno-histochemical labeling of PrP^res deposits (dashed line, black) across 14 brain regions. Although there are some differences, mostly in terms of spongiform lesion intensity, stMI-03, btMI-05, and btMI-09-inoculated animals show highly coincident lesions and PrP^res staining. Animals inoculated with stMI-01 are the only ones with a clearly distinguishable spongiform lesion profile and without detectable PrP^res deposits in all areas analyzed. H&E: Hematoxylin and eosin staining; IHC: Immunohistochemistry.

btMI-05-inoculated animals, showing punctate or granular deposits associated with glia or neurons and in the neuropil. Staining was slightly more intense in the thalamus for stMI-03, and also in mesencephalon, medulla oblongata, and cerebellum for btMI-09. Differences between the three groups were subtle, mainly affecting lesion intensity in some areas, especially for stMI-03. This suggests either infection by a single strain in all three cases or the convergence *in vivo* of the recombinant preparations into a single strain, possibly due to adaptation to the brain homogenate environment in the host or to the selection of a single most favored conformer from a mixture (S6 Fig).

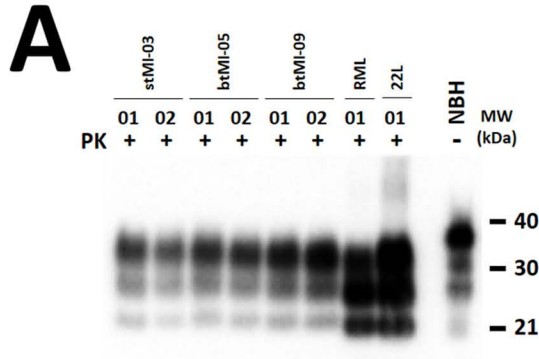

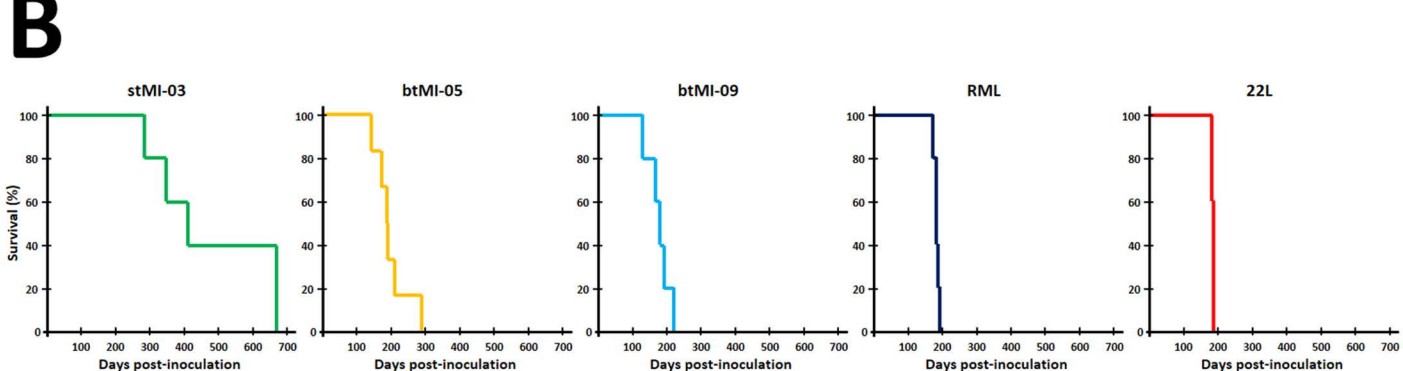

**Fig 5. Biochemical analysis and survival curves of wild-type mice inoculated with spontaneously misfolded PMSA products. A) Biochemical analysis of wild-type brains inoculated with four selected PMSA products**. Brain homogenates from all inoculated animals showing clinical signs of transmissible spongiform encephalopathy were analyzed by proteinase K (PK) digestion, electrophoresis, and Western blot (Sha31, 1:4,000). Results revealed the presence of classical three-banded pattern PrPSc, demonstrating the infectious capacity and *bona fide* nature of the recombinant prions generated spontaneously by PMSA in wild-type animals. The gel shows two representative samples from each group inoculated with the distinct recombinant products (except stMI-01, where all animals remained healthy after 600 dpi and were PrPSc negative) and the brain-derived RML and 22L prions. All the samples are indistinguishable from each other but easily differentiated from the controls. **B) Kaplan-Meier survival curves of wild-type inoculated with PMSA products and classical murine prion strains**. Kaplan-Meier survival curves illustrate the incubation periods following intracerebral inoculations with the distinct PMSA products, RML, and 22L. stMI-03 shows great dispersion in incubation periods, while btMI-05 Dx and btMI-09 Dx exhibit profiles more similar to brain-derived strains RML and 22L, with lower dispersion but longer incubation periods. The extended incubation periods could suggest a need for adaptation or the existence of a transmission barrier between recombinant prions and brain-derived PrPC. Groups showing greater dispersion might indicate stronger barriers due to strain characteristics, mixtures of conformers undergoing slightly different selection process in each animal, or the formation of unstable or somewhat immature conformers. PK: Proteinase K; NBH: Undigested normal brain homogenate; MW: Molecular weight marker.

## Inoculation of recombinant prions in TgVole mouse model demonstrates their interspecies transmissibility and reveals further pathobiological differences between the recombinant prions

To further characterize the spontaneously generated recombinant murine prions and clarify potential strain distinctions, we intracerebrally inoculated them into transgenic mice expressing bank vole I109 PrPC at levels comparable to the wild-type vole [TgVole (1x)], a model useful for strain typing purposes [54,55]. As shown in Table 3, all four recombinant prion strains induced disease in all inoculated animals, albeit with notable differences in incubation periods, statistically significant for most comparisons except for btMI-05 vs. btMI-09 (p-value = 0.349 according to Mann-Whitney U test). However, these differences could be due to titer, ultrastructural arrangement, or a potential transmission barrier between recombinant

**Table 3. Intracerebral inoculation of spontaneously generated PMSA products with distinct electrophoretic migration patterns in TgVole (1x) mice.**

| Strain | Passage | Origin[a] | TgVole (1x) mice | | |
|---|---|---|---|---|---|
| | | | Attack rate | Days post-inoculation ( ± SEM) | PrP^Sc classic pattern (WB) |
| stMI-01 | 1st | rec-MoL108I Dx | 4/4[♯] | 316 ± 48[$] | 4/4 |
| stMI-03 | 1st | rec-MoL108I Dx | 6/6 | 181 ± 8 | 6/6 |
| btMI-05 | 1st | rec-MoL108I Dx | 7/7 | 143 ± 11 | 7/7 |
| btMI-09 | 1st | rec-MoL108I Dx | 6/6 | 123 ± 3 | 6/6 |
| RML | 1st | Brain homogenate | 5/5 | 133 ± 3 | 5/5 |
| 22L | 1st | Brain homogenate | 6/6 | 138 ± 2 | 6/6 |
| | | | | | |
| stMI-01> TgVole (1x) | 2nd | Brain homogenate | 7/7 | 94 ± 2 | 7/7 |
| stMI-03> TgVole (1x) | 2nd | Brain homogenate | 5/5 | 99 ± 4 | 5/5 |
| btMI-05> TgVole (1x) | 2nd | Brain homogenate | 8/8 | 111 ± 5 | 8/8 |
| btMI-09> TgVole (1x) | 2nd | Brain homogenate | 7/7 | 94 ± 1 | 7/7 |
| RML> TgVole (1x) | 2nd | Brain homogenate | 6/6 | 93 ± 3 | 6/6 |
| 22L> TgVole (1x) | 2nd | Brain homogenate | 5/5 | 138 ± 6 | 5/5 |

[a] **Brain homogenate:** Indicates that the seed used as inoculum originates from a 10% brain homogenate in PBS (phosphate-buffered saline) derived from an animal infected with the specified prion strain. The homogenates were further diluted 1:10 in PBS and 20 µl were inoculated intracerebrally in each animal. **rec-MoL108I Dx:** Indicates that the seed used as inoculum originates from the recombinant L108I mouse PrP protein, which was misfolded using the cofactor sulfated dextran, resulting in the indicated prion strain. The PMSA products were diluted 1:10 in PBS and 20 µl of these diluted preparations were inoculated intracerebrally.

[♯] The attack rate was calculated considering only the animals that were euthanized and exhibited a classic PrP^Sc pattern on Western blot (¤) and or by IHC.

[$] The standard error of the mean (SEM) for this group of animals was calculated by including all euthanized animals, encompassing even those that exhibited signs of developing a spontaneous disease.

and brain derived PrP, thus insufficient to determine definitive strain differences. RML and 22L were also inoculated for comparison, showing expected short incubation periods in this widely susceptible model. Despite the overall shorter incubation period likely due to the characteristics of bank vole PrP^C, it is noteworthy that as in previous bioassays, the relative swiftness of each recombinant prion is conserved in this model too, being btMI-09 the fastest, followed by btMI-05, stMI-03 and finally the stMI-01, which as the slowest one is only able to cause disease in overexpressing models (such as the TgMoL108I) or in highly susceptible models such as TgVole (1x). Biochemical analysis of all inoculated mice brains showed the PrP^Sc signature typical of prion disease (S7 Fig) but with mild differences between some animals of the same group, as in the case of stMI-03 and possibly stMI-01.

Upon second passage, incubation periods of the originally recombinant prions were remarkably reduced, converging to times comparable with those of RML and 22L prions (Table 3), and with statistically significant differences only found between stMI-01 and btMI-05-inoculated animals (p-value = 0.023, Mann-Whitney U test). Detailed brain analysis revealed both differences and commonalities between groups. stMI-01 and stMI-03 presented mild spongiosis in the brainstem, with some animals (2/7 and 2/4, respectively) featuring a characteristic intense spongiform lesion focalized in the striatum. Faint punctiform and intracellular PrP^res deposits were detected in the brainstem, with some mice showing plaques in hippocampus and granular-coalescent deposits in the striatum. In contrast, btMI-05 and btMI-09 groups were very homogenous, all animals showed the distinct focalized striatum

focalized spongiosis pattern observed in a few animals of the previous groups. Animals inoculated with btMI-05 were distinguishable from the remaining groups by an intense spongiosis in temporal, piriform, and occipital cortices and moderate spongiform lesions in hippocampus. btMI-09, despite sharing intense striatum spongiosis, presented mostly spared brainstem, neocortex, and mild lesions in the hippocampus (that were absent in stMI-01 and stMI-03). btMI-05-inoculated animals showed granular labeling associated with striatum vacuoles and intense plaque labeling in hippocampus. btMI-09-inoculated animals exhibited fine punctate intraneuronal patterns with plaques primarily in hippocampus (S8 Fig). All four inocula shared the focalized striatum affectation, a generalized finding in btMI-05 and btMI-09 groups but present only in some animals of the other two groups. btMI-05 and btMI-09 were both characterized by presenting abundant plaques in the hippocampus, but at least one animal in groups stMI-01 and stMI-03 also had incipient hippocampal plaques. btMI-05 and btMI-09 were differentiated by lesion intensity in temporal, piriform, and occipital cortices, which was only present in the first. While these results could suggest slightly different strains, the similarity of lesions in some animals across groups could also indicate a single or highly similar strains with varying degrees of severity or subtly different propagation kinetics. Together with the biochemical analysis of the brains, which also showed faint differences between animals inoculated with the same product, the existence of multiple conformers within the same PMSA product cannot be ruled out. These conformers could have been differentially selected and propagated in distinct animal models.

## Dextran sulfate as a specific cofactor: Important for strain properties but not necessary for infectivity in vivo

To explore the role of the cofactor in infectivity and strain characteristics, we propagated the four distinct recombinant mouse prion strains in a PMSA substrate without dextran sulfate, using only conversion buffer (CB, the same buffer used for the preparation of dextran complemented substrates, containing 0.15 M of NaCl and 1% Triton-X-100 diluted in PBS) and recombinant mouse PrP with isoleucine at position 108. We performed four serial 24-h PMSA rounds in quadruplicate, using zirconia-silicate embedded recombinant seeds for the first round and progressively reducing the amount of dextran sulfate in subsequent reactions.

The products of the first PMSA round without dextran sulfate (named stMI-01 CB, stMI-03 CB, btMI-05 CB, and btMI-09 CB) showed notable alterations of their original electrophoretic patterns after PK digestion, electrophoresis, and total protein staining (Fig 6, upper gel). stMI-01 CB, btMI-05 CB, and btMI-09 CB lost the ~16 kDa fragment and showed an intense band of around 10 kDa with small molecular weight fragments of 5 kDa or less. In contrast, stMI-03 CB was characterized by a very intense 16 kDa band with a ladder-like pattern and slightly more intense bands at ~10 and 5 kDa. Nonetheless, given the potential effect of the dextran sulfate on the digestion, as shown in the next section, changes in the electrophoretic pattern could not necessarily indicate a change in strain properties. These dramatic changes in electrophoretic mobility patterns could indicate structural changes in each conformer, potentially altering their biological properties or strain features.

To assess whether the recombinant prions propagated without cofactor had evolved into different strains or lost infectivity, as proposed in previous studies [56], we selected strains stMI-03 CB and btMI-09 CB for further investigation. We chose these two as representative examples because stMI-03 CB showed a unique pattern change, while btMI-09 CB was the fastest among the three strains with similar pattern changes when complemented with dextran sulfate. Replicate 01 of each preparation was selected and further propagated in PMSA for four additional 24-h rounds using 1:100 dilutions between rounds (Fig 6, gels in the bottom).

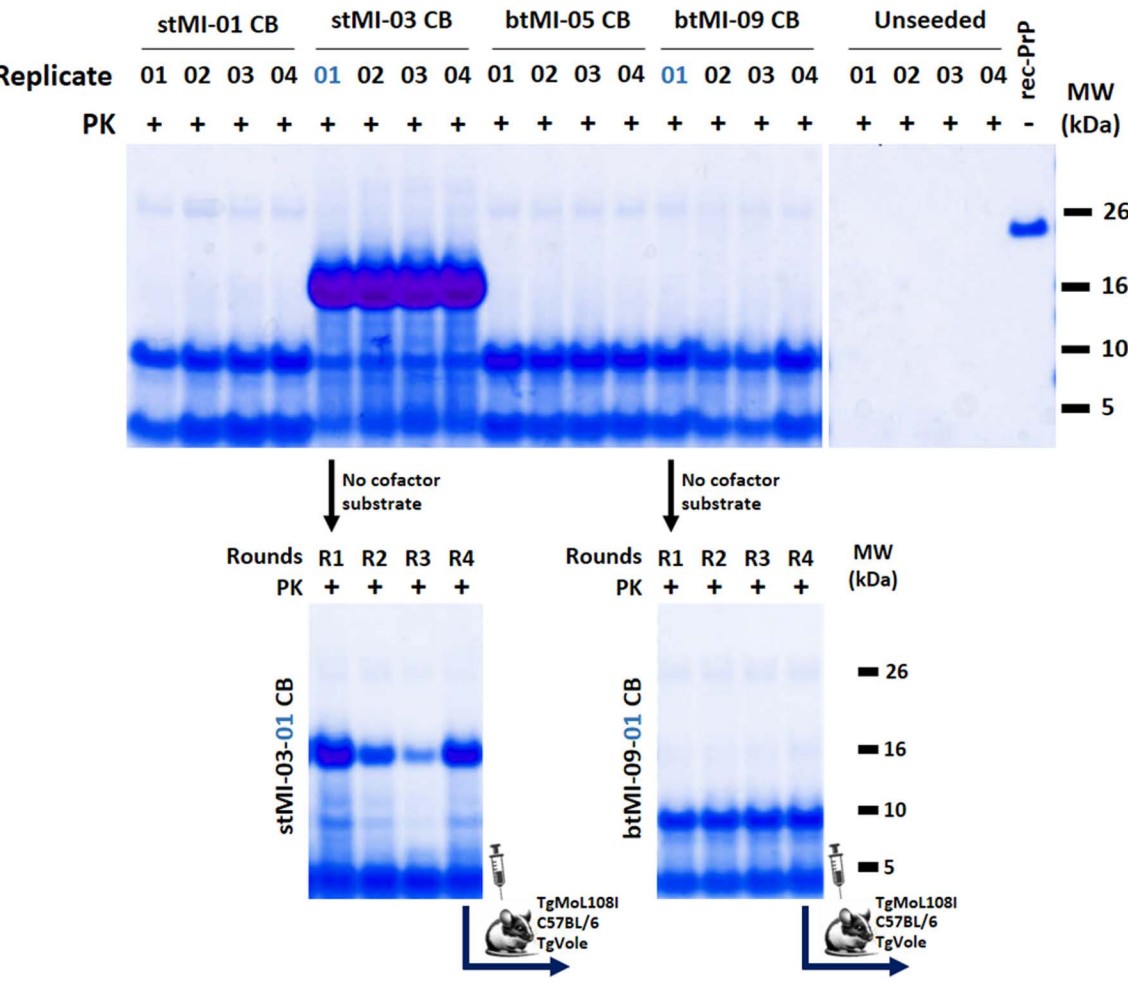

**Fig 6. Adaptation of dextran sulfate-generated recombinant prions to a cofactor-free environment. A) Electrophoretic patterns of recombinant prions propagated without cofactor.** Four recombinant prions underwent four serial PMSA rounds in a substrate containing mouse L108I rec-PrP without dextran sulfate (referred to as CB). The first round used prion-coated zirconia-silicate beads, while subsequent rounds used 1:100 dilutions of previous products. Results were visualized by PK digestion, electrophoresis, and total protein staining. Unseeded control tubes were included to monitor cross-contamination or spontaneous misfolding. The adapted prions show distinct electrophoretic mobility patterns compared to their original dextran sulfate-complemented counterparts. stMI-01 CB, btMI-05 CB, and btMI-09 CB (derived from stMI-01, btMI-05, and btMI-09 generated in presence of dextran sulfate, respectively) converged to a similar profile characterized by the loss of the ~16 kDa fragment present in the originals. In contrast, stMI-03 CB (derived from stMI-03) exhibited a unique profile, distinct from both its original preparation and other adapted products, showing an intense 16 kDa band and a ladder of smaller proteolytic fragments. **B) Electrophoretic pattern conservation during additional PMSA rounds.** Prior to intracerebral inoculations in TgMoL108I, C57BL/6 wild-type mice, and TgVole (1x) models, the adapted products underwent further PMSA propagation to ensure stable biochemical features and eliminate any remaining original seeds. Replicate 01 of stMI-03 CB and btMI-09 CB were used as seeds for four additional 24-h serial PMSA rounds (R1 to R4), demonstrating maintenance of their distinct electrophoretic signatures. The product of the fourth serial round was used for intracerebral inoculations, as illustrated. rec-PrP: Control showing the undigested PMSA substrate containing mouse L108I rec-PrP. MW: Molecular weight marker. The drawing of the mouse shown in the figure was generated using Copilot Designer generative AI tool (Microsoft365).

The products from the fourth round were then inoculated in TgMoL108I intracerebrally for comparison with the original PMSA products spontaneously obtained in presence of dextran sulfate (referred to from now on as stMI-03 dex and btMI-09 dex). Additionally, to gain further insight on strain characterization, stMI-03 CB and btMI-09 CB recombinant products were inoculated intracerebrally in a transgenic mouse model expressing 1x the bank vole PrP

with isoleucine at position 109 [TgVole (1x)] and in wild-type mice (C57BL/6 strain), as done before for stMI-03 dex and btMI-09 dex.

As shown in Table 4, after adaptation to a cofactor-devoid environment, stMI-03 CB was unable to cause clinical disease in wild-type mice after more than 600 dpi, being significantly distinct to stMI-03 dex (p-value = 0.363, Mann-Whitney U test). In the case of TgMoL108I, despite showing also statistically significant differences with the original dextran-complemented preparation (p-value = 0.0008, Mann-Whitney U test) and the btMI-09-derived preparations, given that the incubation period exceeded the time at which the animals of this transgenic line start developing a spontaneous and atypical prion disease, it was also deemed non-infectious, since their biochemical analysis revealed absence of classical three-banded pattern PrP$^{Sc}$. These results could suggest a loss of infectivity due to absence of cofactor. Nonetheless, successful infection of a widely susceptible model such as TgVole (1x), as well as the results obtained with btMI-09 CB, which is infectious in the three models as the original seed complemented with dextran, support an alteration of the strain properties rather than a loss of infectivity. This alteration can respond to a conformational change of the original conformer generated in presence of dextran if a single conformer is assumed, or the selection of some minor conformer, assuming the formation of distinct conformer mixes in the original preparation and being the latter supported by previous bioassays with the original PMSA products and TEM. Biochemical analysis of the brains of the animals showing neurological disease signs confirms the induction of a *bona fide* prion disease, with classical, three-banded PrP$^{Sc}$ detected in all cases (S9 Fig). Therefore, both stMI-03 CB and btMI-09 CB keep their infectivity when inoculated in suitable models, confirming that a polyanionic cofactor such as the dextran sulfate used in this case is not necessary for the recombinant prions to be infectious *in vivo*.

Histopathological analysis (S10A Fig) of the brains of the wild-type animals inoculated with stMI-03 CB showed no PrP$^{res}$ and very mild spongiform changes in brainstem, likely due to the advanced age of the animals, further confirming the unsuccessful infection. For TgMoL108I, given the absence of classical PrP$^{Sc}$ by Western blot, no further analysis was carried out, as the lesions found would respond to either spontaneously developed atypical disease or a mixture of induced and spontaneous disease. The analysis of the stMI-03 CB-inoculated TgVole (1x) brains revealed heterogeneity. This is likely due to interference from a subclinical and atypical spontaneous disease. This disease has been previously observed in uninoculated animals at advanced ages and in TgVole overexpressing 4x the

**Table 4. Intracerebral inoculation of PMSA products, generated spontaneously with or without cofactors, in TgMol108I, C57BL/6, and TgVole (1x) mice.**

| Strain | Cofactor presence | TgMoL108I mice | | | C57BL/6 | | | TgVole (1x) mice | | |
|---|---|---|---|---|---|---|---|---|---|---|
| | | Attack rate | Days post-inoculation (± SEM) | PrP$^{Sc}$ classic pattern (WB) | Attack rate | Days post-inoculation (± SEM) | PrP$^{Sc}$ classic pattern (WB) | Attack rate | Days post-inoculation (± SEM) | PrP$^{Sc}$ classic pattern (WB) |
| **stMI-03 dex** | Yes | 6/6 | 207 ± 11 | 6/6 | 3/5[ƒ] | 472 ± 81 | 3/5 | 6/6 | 181 ± 8 | 6/6 |
| **stMI-03 CB**[a] | No | 0/8 | > 250[¤] | 0/8 | 0/6 | > 600 | 0/6 | 5/5 | 462 ± 15 | 5/5 |
| **btMI-09 dex** | Yes | 5/5 | 158 ± 5 | 5/5 | 5/5 | 174 ± 15[$] | 5/5 | 6/6 | 123 ± 3 | 6/6 |
| **btMI-09 CB**[a] | No | 8/8 | 170 ± 4 | 8/8 | 5/5 | 424 ± 33 | 5/5 | 5/5 | 116 ± 4 | 5/5 |

[a] The prion strain was serially propagated using PMSA without the addition of cofactors (see Fig 6).

[¤] TgMoL108I mice spontaneously develop a TSE at 339 ± 9 days. Therefore, if a prion strain exhibits a longer incubation time in this model, its actual infectivity cannot be accurately assessed.

[ƒ] The attack rate was calculated considering only the animals that were euthanized and exhibited a classic PrP$^{Sc}$ pattern on Western blot and/or by IHC.

[$] In this group, 2 animals developed intercurrent diseases at 137 dpi, and were excluded from the calculations of attack rate and incubation periods.

same protein, where it develops much earlier [57,58]. The characteristics of this disease are well known to us. This allowed separate assessments of the histopathological features corresponding to the spontaneous disease from the lesions caused by the recombinant prion. Approximately half of the animals showed notable spongiform lesions in hippocampus, cerebellar cortex, and striatum indicative of spontaneous atypical disease, while the other half presented moderate spongiosis in brainstem and cerebellar cortex, compatible with the induced prion disease. Localization and pattern of PrP$^{res}$ through immunohistochemistry was also heterogeneous in this group. Some animals of the group showed intraneuronal punctate labeling and mild labeling in the neuropil (classical pattern), while others presented plaques in striatum, and very mild granular staining in the alveus of the hippocampus (atypical pattern). This mixture of phenotypes makes it extraordinarily challenging to address if stMI-03 CB presents the same strain features as stMI-03 dex. However, considering only those animals in which the influence of the spontaneous disease seems minimal, spongiform lesion profiles are quite different, with severe affectation of the striatum in the case of the original strain that is mild in the CB-passaged preparation, suggesting an alteration of the strain properties of stMI-03 dex upon propagation in a cofactor-devoid environment.

The study on animals inoculated with btMI-09 CB confirms its capacity to induce a *bona fide* prion disease in TgMoL108I, in which no statistically significant differences were found in comparison to btMI-09 dex (p-value = 0.118, Mann-Whitney U test); wild-type mice, that revealed significant differences between the two groups (p-value = 0.011); and TgVole (1x), in which the difference resulted non-significant according to Tukey's post hoc test (p-value = 0.884); showing differential features in each model (S10A Fig). TgMoL108I brains show very mild spongiform lesions localized in brainstem and cerebellum, with evident PrP$^{res}$ plaques in the white matter of the cerebellum, and intraneuronal punctate labeling in brainstem and occasionally in cortices and hippocampus. Inoculation of btMI-09 CB in wild-type mice resulted in intense spongiform lesions in brainstem and cortices. PrP$^{res}$ deposits were undetectable in some animals while in others, they were characterized by very low labeling of granular intraneuronal and neuropil deposits. Comparison of the spongiform lesion profile in wild-type mice with that induced by the original btMI-09 dex in the same animal model (S10B Fig) showed highly similar patterns, suggesting that, in contrast to stMI-03, propagation in a cofactor-devoid environment did not significantly alter the original strain.

TgVole (1x) brains showed very mild spongiform lesion in brainstem and, exceptionally, in the hippocampus of a single animal. Similarly, PrP$^{res}$ labeling was intraneuronal and very mild (only detectable with high magnification) with a few aggregates detected in few animals. The PrP$^{res}$ was mainly restricted to the brainstem, although in some cases it could be detected also in cerebellar cortex and hippocampus. In contrast to the results in wild-type, comparison of the spongiform lesion profiles of btMI-09 CB and the original dextran-complemented strain in TgVole (1x) (S10C Fig), despite showing highly coincidental lesions in most areas, are distinguished by the severe lesion in striatum found only in the dextran-complemented prion. This severe focal affectation of the striatum, however, was found with all dextran-complemented recombinant strains upon inoculation in TgVole (1x), while absent in the rest of the models inoculated with the same preparations. Thus, it is possible that this focalized striatum affectation in TgVole (1x) is a characteristic of the model in combination with any dextran-complemented strain and may not necessarily indicate conformational differences between btMI-09 CB and btMI-09 dex. The alteration in electrophoretic mobility pattern and increased incubation period in wild-type mice suggests conformational changes may have occurred.

In the light of these results, it is clear that the dextran sulfate is not necessary to cause infection *in vivo*. However, it seems able to influence the biochemical and biological properties of prions in a strain-dependent manner, being in this case apparently more influential for stMI-03 than for btMI-09.

## Analysis of the pathobiological features of recombinant prion btMI-09 after switching cofactor presence in the propagation environment indicates strain-dependent effect

We propagated the cofactor-devoid product (btMI-09 CB) in the original substrate containing dextran sulfate and mouse L108I rec-PrP. btMI-09 CB was first loaded onto zirconia-silicate beads for use as seeds in PMSA. Three 24-h serial PMSA rounds were performed, starting with prion-loaded beads and using 1:10 dilutions between rounds. The electrophoretic mobility pattern of the resulting strain, named btMI-09 dex$^2$, was then analyzed. Finally, btMI-09 dex$^2$ was re-adapted to a dextran-devoid environment through 4 serial PMSA rounds, starting from btMI-09 dex$^2$-coated zirconia-silicate beads and using 1:10 dilutions between rounds. As shown in Fig 7A, adaptation of btMI-09 CB back to a dextran sulfate-containing environment (btMI-09 dex$^2$) led to the recovery of the ~16 kDa band, resulting in an electrophoretic mobility pattern similar to the original btMI-09 dex. However, minor differences were observed in the proportion of fragments, with the 16 kDa band more intense in the original preparation compared to lower molecular weight fragments, while btMI-09 dex$^2$ showed similar relative intensity among PK-resistant fragments. Further adaptation of btMI-09 dex$^2$ to a cofactor-devoid environment (btMI-09 CB$^2$) resulted in the loss of the 16 kDa fragment, similar to the original btMI-09 dex to btMI-09 CB transition. However, since dextran sulfate, if strongly associated to prion fibers, could affect proteinase K accessibility or activity, changes in electrophoretic mobility as indicators of potential strain differences should be evaluated carefully. In fact, the digestion of btMI-09 CB and btMI-09 CB$^2$ in the presence of dextran sulfate (Fig 7C) revealed that this compound hinders the digestion of the 16 kDa fragment, detectable only when the digestion was done in the presence of this molecule.

To assess whether the slight differences in electrophoretic mobility patterns corresponded to a recovery of the original strains or the emergence of distinct ones, we intracerebrally inoculated btMI-09 dex$^2$ and btMI-09 CB$^2$ into TgMoL108I mice for comparison with previously inoculated btMI-09 dex and btMI-09 CB (Table 5). Five animals were inoculated with btMI-09 dex$^2$, all succumbing to prion disease with a mean incubation period of 259 ± 28 dpi. Similarly, six animals were inoculated with btMI-09 CB$^2$, showing 100% attack rates and a mean incubation period of 189 ± 10 dpi. In both cases, a prolongation of the incubation periods with respect to btMI-09 dex and btMI-09 CB was detected, which could also indicate some alteration of strain properties although it was only statistically significant when comparing btMI-09 dex and btMI-09 CB$^2$ (p-value = 0.035, Mann-Whitney U test). However, as shown in Fig 7B, the PrP$^{res}$ detected in the brains of animals with neurological signs exhibited a classical three-banded pattern, indistinguishable from that observed in animals inoculated with btMI-09 dex and btMI-09 CB preparations. Further analysis of the animals inoculated with btMI-09 dex$^2$ and btMI-09 CB$^2$ in comparison to those inoculated with btMI-09 dex and btMI-09 CB (S11A Fig), reveals mild spongiform lesions in btMI-09 dex$^2$-inoculated animals mostly in thalamus, mesencephalon and cerebellar cortex. These findings are reminiscent of those found in the previous btMI-09-CB preparation. In addition, this group shows evident plaques in the cerebellar cortex white matter with an immunolabeling similar to that observed in the previous group. However, some differences

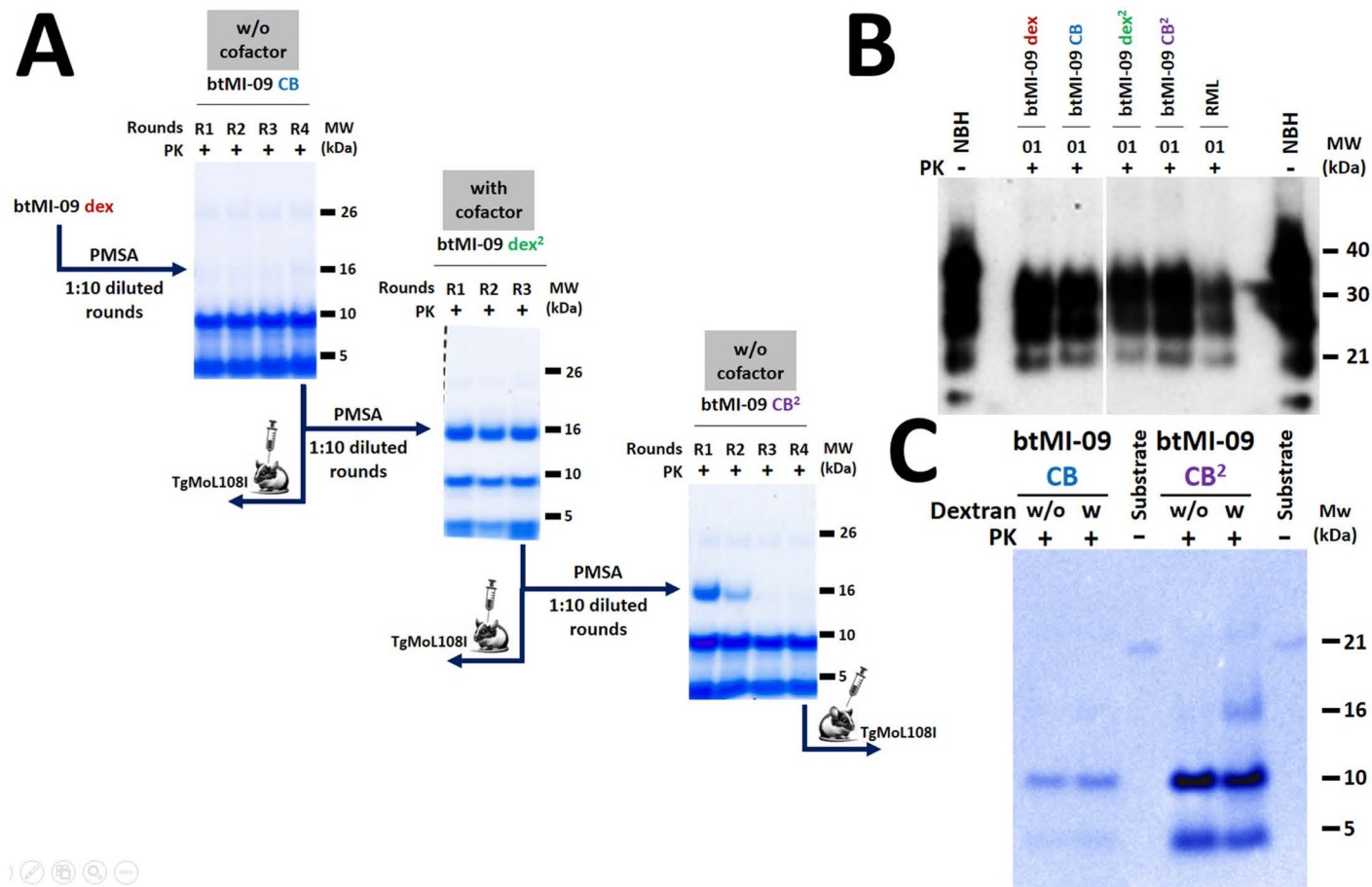

**Fig 7. Propagation of btMI-09 CB in dextran sulfate-complemented environment and back in a cofactor-devoid substrate and biochemical analysis of TgMoL108I brains inoculated with all the resulting inocula. A) Electrophoretic patterns of recombinant prions propagated without cofactor (btMI-09 CB), then with dextran sulfate as cofactor (btMI-09 dex²) and back without cofactor (btMI-09 CB²).** btMI-09 dex recombinant prions underwent four serial PMSA rounds (R1-R4) in a substrate containing mouse L108I rec-PrP without dextran sulfate (referred to as CB). Results were visualized by PK digestion, electrophoresis, and total protein staining. After stabilization of btMI-09 CB, this procedure was repeated: first using a dextran sulfate-complemented substrates (only 3 PMSA rounds were performed prior to coating of zirconia-silicate beads), resulting in btMI-09 dex² PMSA product and then again in a cofactor-devoid environment, giving rise to btMI-09 CB². The dextran sulfate-complemented prions, the original btMI-09 dex and btMI-09 dex², show almost indistinguishable electrophoretic mobility patterns, with subtle differences in low molecular weight fragment intensity. Analogously btMI-09 CB and btMI-09 CB² share a similar profile characterized by the loss of the ~16 kDa fragment present in the dextran sulfate-propagated products, also presenting subtle differences in the relative intensity of the low molecular weight fragments. **B) Biochemical analysis of TgMoL108I mice brains inoculated with the four btMI-09 preparations adapted to dextran sulfate-complemented and cofactor-devoid substrates**. Brain homogenates from all inoculated animals showing clinical signs of transmissible spongiform encephalopathy were analyzed by proteinase K (PK) digestion, electrophoresis, and Western blot (Sha31, 1:4,000). Results revealed the presence of classical three-banded pattern PrP^Sc, demonstrating the infectious capacity and *bona fide* nature of the four products btMI-09 dex, CB, dex² and CB², after sequential adaptation to the two distinct propagation environments. The gel shows a representative example from each group inoculated with the distinct recombinant products and one of the brain-derived RML as control of a classical prion strain. PrP^Sc from all animals inoculated with the different versions of btMI-09 are indistinguishable from each other. **C) Digestion of preparations adapted to cofactor-free environment in the presence and absence of dextran sulfate during digestion.** To evaluate the potential interfering effects of sulfated dextran on proteinase K digestion patterns, btMI-09 CB and btMI-09 CB² were incubated during 1 h in the absence (w/o) or presence (w) of dextran sulfate, showing that the detection of the 16 kDa proteolytic fragment depends on the presence of dextran in the reaction, likely due to its strong association to fibers. PK: Proteinase K; NBH: Undigested normal brain homogenate; MW: Molecular weight marker. The drawing of the mouse shown in the figure was generated using Copilot Designer generative AI tool (Microsoft365).

were also noticed, mainly in terms of cortical spongiform lesions, more evident in the btMI-09 dex², and the presence of PrP^res in hippocampus, found mainly in the btMI-09 CB-inoculated animals. Upon the secondary adaptation to a cofactor-devoid environment, btMI-09 CB² inoculated mice, also show very mild spongiform changes, localized in the

**Table 5. Intracerebral inoculation of the btMI-09 strain following serial passages of PMSA with or without cofactor in TgMol108I, C57BL/6 and TgVole (1x) mice.**

| Strain[a] | Cofactor presence | TgMoL108I mice | | |
|---|---|---|---|---|
| | | Attack rate | Days post-inoculation (± SEM) | PrP^Sc classic pattern (WB) |
| btMI-09 dex | Yes | 5/5 | 158 ± 5 | 5/5 |
| btMI-09 CB | No | 8/8 | 170 ± 4 | 8/8 |
| btMI-09 dex$^2$ | Yes | 5/5 | 259 ± 28 | 5/5 |
| btMI-09 CB$^2$ | No | 6/6 | 189 ± 10 | 6/6 |

[a] The prion strain btMI-09, initially generated using the dextran sulfate cofactor (btMI-09 dex), was serially propagated using PMSA without cofactors, resulting in btMI-09 CB. This strain was subsequently serially propagated again using PMSA with dextran sulfate (btMI-09 dex$^2$). Finally, the misfolded protein underwent other serial rounds of propagation without cofactors, yielding btMI-09 CB$^2$ (see Fig 7).

thalamus and also in the hippocampus in a few animals. Plaques were observed again in the white matter of the cerebellar cortex and to some extent in brainstem and hippocampus, with the same granular pattern observed in previous groups. Therefore, the four preparations share similar lesion profiles and PrP^res labeling patterns in TgMoL108I (S11B Fig), suggesting no significant alteration of strain properties related to the cofactor in the propagation environment. However, minor differences were noted, aligning with the slightly distinct electrophoretic mobility profiles and incubation periods. These minor alterations appeared more pronounced in specific animal models, such as the TgVole (1x), where the affectation of the striatum differed noticeably between dextran-complemented and -devoid preparations. These observations suggest that the presence of polyanionic cofactors in the propagation environment may influence the pathobiological properties of prions, likely in a strain-dependent manner.

Finally, after adapting the dextran sulfate-generated recombinant prions to a cofactor free-environment and subsequently returning them to the original dextran sulfate-containing substrate, their ultrastructural arrangement was characterized using transmission electron microscopy and negative staining. This analysis was performed in comparison to the original recombinant prions generated in the presence of dextran sulfate (S12 Fig). For stMI-03, despite the differences observed in the electrophoretic mobility pattern and biological properties between dextran-complemented and dextran-free preparations, we could not find any notable differences in terms of ultrastructural arrangements. Similarly, the four btMI-09 preparations were indistinguishable, which aligns with the histopathological findings. Additionally, the electrophoretic mobility patterns of these preparations after secondary transmission to wild-type mice (both, secondary transmissions from TgMoL108I to wild type, and those performed directly in wild-type mice) were compared with all the PMSA products that resulted infectious in C57BL/6 mice, revealing no differences among recombinant preparations transmitted to wild-type mice and thus supporting that btMI-09 dex and btMI-09 CB could be identical (S13 Fig). Moreover, this figure shows the convergence of electrophoretic mobility profiles after secondary transmission to wild type mice, despite in some cases differences were detected in first passages to either TgMoL108I (Fig 3) or C57BL/6 (Fig 5). These findings indicate that most of their pathobiological characteristics are conserved upon propagation in distinct environments. Differences were only detectable upon inoculation in TgVole (1x) mice.

## Spontaneous misfolding of murine L108I recombinant PrP in absence of cofactor gives rise to bona fide prions

Our previous results indicate that infectious recombinant prions can be propagated in a cofactor-devoid environment, maintaining their capacity to induce TSE *in vivo*, albeit in a potentially strain-dependent manner. These findings prompted us to attempt spontaneous misfolding of mouse L108I rec-PrP by PMSA in the absence of dextran sulfate cofactor. This experiment aimed to determine whether this polyanionic cofactor is necessary for *bona fide* recombinant prion misfolding or if it merely acts as an inductor facilitating the formation of specific prion conformers or strains without being indispensable for obtaining infectious prions.

Unlike the dextran-complemented substrate, rec-PrP$^{res}$ generation was inconsistent across different experiments and replicates. In many cases, after the initial 24 h of PMSA, we detected protease-resistant products characterized by low molecular weight fragments (~10 kDa and < 5 kDa). In subsequent rounds, these evolved into the electrophoretic pattern associated with recombinant *bona fide* prions, characterized by a 16 kDa fragment.

We selected results from two independent experiments that showed the expected pattern in the second serial PMSA round for further propagation and characterization. No other distinct profiles were detected. Fig 8A demonstrates the stable propagation of the electrophoretic mobility profiles of the two selected PMSA products, named MoL108I-CB-01 and MoL108I-CB-02.

These spontaneously generated misfolded and PK-resistant PMSA products were intracerebrally inoculated in TgMoL108I and wild-type mice to assess their infectivity *in vivo*. As shown in Table 6, both preparations showed 100% attack rates and similar incubation periods in the case of TgMoL108I, being the difference statistically no significant between them (p-value = 0.682, Mann-Whitney U test) (Table 1). In wild-type animals, MoL108I-CB-01 also showed complete attack rates and, in this case, statistically significant differences with stMI-03 CB (p-value = 0.001) and btMI-09 CB (p-value = 0.011) preparations. Upon secondary transmission to the same model, incubation times were significantly reduced, reaching 136 ± 3 dpi and showing no statistically significant differences with respect to btMI-09 dex or btMI-09 CB inoculated in the same model, demonstrating also the transmissibility of these recombinant prion generated in absence of cofactor and an incubation period in line with other brain-derived murine prion strains. Therefore, both spontaneously generated recombinant prions in the absence of cofactor are definitively infectious *in vivo*, showing an incubation period upon adaptation to wild-type mice like that of the recombinant strains generated with dextran sulfate (Table 2), and faster than the btMI-09 CB.

Table 6. Intracerebral inoculation of PMSA products, generated spontaneously without cofactors, in TgMoL108I and C57BL/6 mice.

| Strain | Cofactor presence | TgMoL108I mice | | | C57BL/6 | | |
|---|---|---|---|---|---|---|---|
| | | Attack rate | Days post-inoculation (± SEM) | PrP$^{Sc}$ classic pattern (WB) | Attack rate | Days post-inoculation (± SEM) | PrP$^{Sc}$ classic pattern (WB) |
| **MoL108I-CB-01**[a] | **No** | 5/5 | 189 ± 8 | 5/5 | 5/5 | 251 ± 10 | 5/5 |
| **MoL108I-CB-02**[a] | **No** | 7/7 | 180 ± 8 | 7/7 | ongoing/6 | > 75 (ongoing) | ongoing |
| **MoL108I-CB-01 > C57BL/6**[b] | – | ND | ND | ND | 6/6 | 136 ± 3 | 6/6 |

[a] The prion strains were spontaneously generated and propagated in the absence of cofactors (see Fig 8).

[b] Transmission to C57BL/6 animals of MoL108I-CB-02 is currently ongoing, the animals not showing any clinical sign at the time of submission (75 dpi).

These results confirm that the cofactor is not necessary for the formation of *bona fide* recombinant prions *in vitro*. Biochemical and anatomopathological analyses of the inoculated animals' brains revealed the characteristic three-banded PrP$^{Sc}$ pattern (Fig 8B), along with the presence of deposits and spongiform lesions (Fig 8C). These findings clearly demonstrate that the recombinant prions generated in the absence of a cofactor are as infectious *in vivo* as those produced in the presence of dextran sulfate, although they may exhibit distinct strain features. Indeed, although lesion profiling was not optimal due to fixed tissue damage or missing brain areas that precluded a more thorough analysis, TgMoL108I brains inoculated with MoL108I-CB-01 and MoL108I-CB-02 showed mild spongiform changes in the brainstem, particularly in the thalamus, and in some cases, the hippocampus. Plaques were observed in the white matter of the cerebellum, with granular punctate labeling also noted in the hippocampus and cortical regions for MoL108I-CB-01, and in the brainstem, hippocampus, and occasionally in other regions for MoL108I-CB-02. Therefore, both preparations are quite similar and may represent the same strain, resembling the btMI-09 CB cofactor-devoid preparation. Additionally, lesion profile of MoL108I-CB-01-inoculated wild-type mice, despite incomplete due to suboptimal sampling mostly in the cerebellar nuclei, show clear spongiform changes in the brainstem and the thalamus, and a very faint punctate intraneuronal and intraglial PrP$^{res}$ labelling, detectable also in neuropil. This profile is reminiscent of those found in wild-type mice inoculated with TgMoL108I passaged, dextran-complemented prions stMI-03, btMI-05 and btMI-09 (Fig 4).

The characterization of their ultrastructural arrangement by TEM (S14 Fig) did not reveal any dramatic differences from the previous PMSA products obtained spontaneously in PMSA in the presence of dextran sulfate. The main distinction was an apparently higher number of twisted fibers with clearly detectable torsions and, in a few cases, smaller half-pitches. Lateral clustering of fibers seemed also less frequent than in the previous preparations, as did the number of curved fibrils, although all previously identified structural elements were conserved. Both MoL108I-CB-01 and MoL108I-CB-02 preparations were indistinguishable from each other, in line with their biochemical and histopathological features.

In conclusion, these results further support the idea that polyanionic cofactors are not necessary for the successful formation of prions *in vitro* or for recombinant prion infectivity *in vivo*. However, they could play a relevant role in determining conformational differences that may result in the formation of distinct strains or conformer mixtures.

## Discussion

Spontaneous PrP$^{C}$ to PrP$^{Sc}$ misfolding is the main pathogenic event underlying the most common prion disease in humans, sporadic Creutzfeldt-Jakob disease (sCJD) [59]. Despite being triggered in specific PRNP variants through yet unknown mechanisms, it could also be considered the cause of genetic prion diseases. Spontaneous misfolding of a wild-type PrP variant, lacking any disease-associated alteration, is an extremely rare and possibly stochastic occurrence. sCJD is estimated to cause the death of 1–2 persons per million per year, with no significant differences across countries [59]. The molecular-level occurrence of this event remains unknown given the difficulty of modeling it both *in vivo* and *in vitro*. The search for genetic or environmental factors related to sCJD has not yet provided significant results explaining this rare phenomenon, despite a recent genome-wide association study detecting promising PrP-independent genetic risk factors [60]. The finding that polyanionic molecules such as RNA could promote spontaneous wild-type PrP misfolding *in vitro* [36], the observation that shortening heparan sulfate delays prion propagation *in vivo* [61], together with the recent discovery of GAL3ST1 (an enzyme that sulfates sphingolipids to make sulfatides) [60], suggest that negatively charged biomolecules, such as polyanions, could be important players in spontaneous prion misfolding.

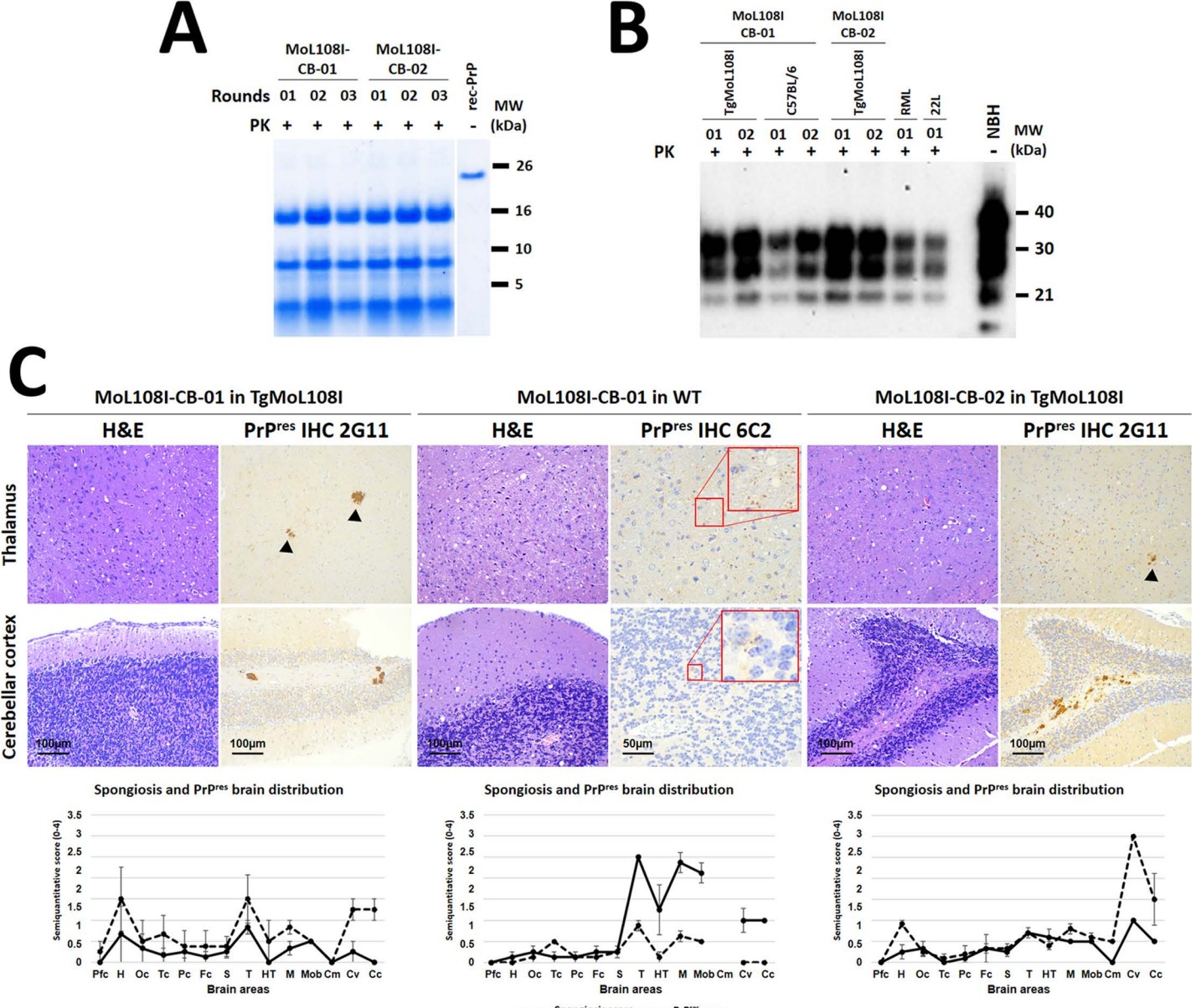

**Fig 8. Characterization of spontaneously misfolded recombinant mouse L108I PrP in PMSA using cofactor -devoid substrate. A) Electrophoretic mobility profile and stable propagation *in vitro* of PMSA products MoL108I-CB-01 and MoL108I-CB-02.** Two PMSA products, MoL108I-CB-01 and MoL108I-CB-02, were generated via spontaneous misfolding of mouse L108I rec-PrP in a dextran sulfate-free substrate (CB). PK-resistant rec-PrP with the expected ~16 kDa fragment characteristic of recombinant prions was detected after the second 24-hour PMSA round and stably propagated for three additional rounds (1:10 seed dilution). PK digestion, electrophoresis, and total protein staining revealed indistinguishable electrophoretic profiles between the two products, consistent across rounds. An undigested CB substrate was included for size comparison. **B) Biochemical analysis of TgMoL108I mice brains inoculated with MoL108I-CB-01 and MoL108I-CB-02 preparations and wild-type mice brains inoculated with MoL108I-CB-01**. Brain homogenates from TgMoL108I and wild-type mice inoculated with MoL108I-CB-01 and MoL108I-CB-02 were analyzed via PK digestion, electrophoresis, and Western blot (Sha31, 1:4,000). All inoculated animals showing transmissible spongiform encephalopathy (TSE) symptoms exhibited the classical three-banded PrP$^{Sc}$ pattern, confirming the infectious capacity and bona fide prion nature of the PMSA products. PrP$^{Sc}$ profiles were indistinguishable across groups and comparable to brain-derived murine prion strains RML and 22L. PK: Proteinase K; NBH: Undigested normal brain homogenate; MW: Molecular weight marker. **C) Brain lesion and PrP$^{res}$ deposit distribution of the TgMoL108I inoculated MoL108I-CB-01 and MoL108I-CB-02 and wild-type mice inoculated with MoL108I-CB-01.** Histopathological analysis revealed distinct lesion and PrP$^{res}$ deposition profiles. TgMoL108I mice inoculated with MoL108I-CB-01 and MoL108I-CB-02 exhibited mild spongiform changes and PrP$^{res}$ deposits primarily in the brainstem (thalamus), hippocampus, and cerebellar cortex (indicated by black arrowheads). PrP$^{res}$-positive plaques in the cerebellar cortex white matter resembled those observed with dextran sulfate-supplemented PMSA products (S11 Fig), suggesting similar strain features. In contrast, wild-type mice inoculated with MoL108I-CB-01 displayed a unique profile, similar to TgMoL108I-passaged stMI-03, btMI-05, and btMI-09 inocula (Fig 4). Spongiform lesions were prominent in the thalamus and cerebellar cortex, with intraneuronal and punctate neuropil PrPres deposits labeled with 6C2 (1:1,000) in the same regions. H&E: Hematoxylin and eosin staining; IHC: Immunohistochemistry.

Our previous research on *in vitro* prion misfolding and propagation, grounded in work demonstrating that spontaneous rec-PrP misfolding could be achieved *in vitro* with minimal components [20,31], led us to develop the Protein Misfolding Shaking Amplification (PMSA) method, a variant of previous amyloid seeding assays [26,27]. This method allows swift and consistent spontaneous misfolding of recombinant bank vole PrP using only dextran sulfate as polyanionic cofactor and glass beads [45]. Although further developments have allowed the use of this method for the spontaneous misfolding of hundreds of mammalian PrP variants [44], the polymorphic variant 109I of bank vole PrP seems to confer enhanced susceptibility to spontaneous misfolding, in agreement with its effect *in vivo* [18]. This effect of isoleucine in the equivalent position to the polymorphism in bank vole is confirmed by our results. Among all possible amino acids at position 108 of mouse PrP, L108I murine rec-PrP was the only variant with spontaneously misfolded rec-PrPres detectable after 4 h of PMSA in 100% of replicates. Although other substitutions in the same position also enhanced the spontaneous misfolding ability of mouse rec-PrP, substitution by isoleucine exerted the strongest effect for unknown reasons. Since the mid-2000s, bank voles have been acknowledged among other rodent models as a highly susceptible and extraordinarily fast model for a diversity of prion strains from multiple species. Despite the bank vole M109 PrP polymorphic variant showing similar enhanced susceptibility to classical prion strains as the I109 variant, they exhibit strikingly different behaviors in terms of susceptibility to atypical prion diseases such as GSS or Nor98, and in terms of spontaneous prion disease development, greatly favored by the isoleucine [62]. Residues N155 and N170 from bank vole have been underscored as the most influential in facilitating classical prion strain propagation, whereas I109 appeared to reduce conversion efficiency compared to the M109 variant [63]. However, the presence of these two residues in other species such as hamster PrP indicates they are not solely responsible for the enhanced susceptibility to interspecies prion transmission, with further studies pointing towards the specific composition of the β2-α2 loop of bank voles [64]. In this regard, the latest study on the molecular determinants of the susceptibility of bank voles (M109 variant) to prions from mouse and hamster, highlights the role of residues V112, I139 and M205 apart from those detected previously [65]. In contrast, it is well established that residue I109 is the most influential for the extraordinary spontaneous misfolding proneness of bank vole PrP [18,66], which allowed us to develop *in vitro* propagation methods to study spontaneous prion misfolding [39,45]. Additionally, we have demonstrated that its effect can be transferable to other species *in vivo*, such as sheep, in which the expression of an M112I variant, interestingly also a natural polymorphism in this species, led to the spontaneous development of an atypical and transmissible prion disease [50], and in the present study to mouse PrP. However, whether this effect depends solely on the presence of an isoleucine in that position remains elusive, as does the molecular mechanism by which this residue dramatically alters the spontaneous misfolding propensity. Apart from bank vole and sheep PrP, there are other six mammalian PrP variants from distinct species (Przewalski's horse or takhi, skunk, ferret, Western spotted skunk, honey badger and four-toed hedgehog) that present an isoleucine in the equivalent position sequenced so far. The *in vitro* evaluation of their spontaneous misfolding proneness by PMSA reveals a variable capacity for this process (see PrPdex webpage), suggesting the presence of this isoleucine alone may be insufficient to confer this capacity to the aforementioned PrP [44]. However, further analysis of the sequences of these PrP variants could shed light on the complex interactions required in addition to this specific amino acid at said position. Recent studies point towards C-terminal residues of bank vole, specifically E227 and S230, as primarily responsible for its particular behavior, independently of I109 [67,68]. This contradicts our findings with sheep and mouse PrP, which lack those C-terminal residues. Nonetheless, it is clear that this I109 residue confers unique properties to bank vole

PrP in terms of spontaneous and induced misfolding. Apart from greatly facilitating both processes *in vivo* and *in vitro*, it favors the formation and propagation of atypical prions. This has been demonstrated by the spontaneous disease developed in overexpressing TgVole mice [57,58], when introducing human disease-associated mutations in this species [69], and through comparative inoculation of atypical ovine prions in M109 and I109 bank voles [70]. Further investigations into the mechanisms by which this residue leads to these unique effects on PrP misfolding could be highly relevant to deciphering prion conversion at a molecular level. Along this line, the recently solved high resolution structures of distinct rodent prion strains from hamster and mouse prions [4–7,9], which differ from bank vole PrP by only a few amino acids, show that residues equivalent to 109, 112, and 139 are part of highly packed hydrophobic stretches critical for prion fiber stability. Therefore, alterations in those residues, already proven to be relevant by previous studies, could strongly influence the formation of these critical hydrophobic cores and the fitting of non-homologous PrP$^C$ in said positions. According to our results, isoleucine would be favorable in these positions.

Interestingly, one of the aforementioned studies also revealed the capacity of bank vole I109 PrP to propagate classical PrP$^{Sc}$ conformers emerging from atypical scrapie isolates. This suggests a higher promiscuity for the propagation of distinct conformers compared to the M109 counterpart, which is unable to adopt the conformation characteristic of atypical prions [70]. This finding aligns with our detection of distinct conformers in the preparations, a diversity that could be related to the universal prion acceptor nature of bank vole PrP. This would indicate its capacity to adopt an enormous variety of prion conformations. The existence of conformer or strain mixtures in isolates derived from prion-affected animals and patients is already a well-known phenomenon in the field [12,71]. Thus, it is not unexpected that they could also form in a cell-free system such as the PMSA. In agreement with the conformational selection model [72] and the demonstration that prions could exist as clouds of multiple conformers or quasi-species populations [73], our results indicate that the PMSA products generated spontaneously may also represent heterogeneous mixtures of distinct conformers. This potential mixture that would require further serial *in vivo* transmissions for full stabilization, among other possible reasons, could explain the apparently lower infectivity titer of the recombinant preparations, in which there could have been up to 10 to 50-fold more of misfolded PrP than in the isolates used as control of brain-derived prions, 22L and RML. Although at a molecular level prion strains have been hypothesized to correspond to specific PrP$^{Sc}$ conformations, and the first handful of high-resolution prion structures have indeed demonstrated variations between distinct classical prion strains [7,9], researchers in the field still use an operational definition of prion strains. This definition is based on their capacity to cause a heritable phenotype of disease under specific conditions such as titer, inoculation route, and amino acid sequences of donor and host [74]. Some biochemical differences could be hinted at for our four selected PMSA preparations, indicating potential strain differences, mainly for stMI-03. However, upon inoculation in different mouse models, similarities were underscored in some cases, while differences were observed in others, hampering robust strain typing or clear distinctions between the PMSA products. Inoculation first in a model bearing homologous PrP$^C$ (TgMoL108I) and then in wild-type mice, or even directly in wild-type mice indicated that stMI-01 was clearly different from the rest, unable to efficiently propagate in this model. It also revealed a differential behavior for stMI-03 despite being similar to the others in terms of lesion profile. Upon the second passage in wild-type, however, stMI-03 converged with btMI-05 and btMI-09. Inter-species transmission to TgVole highlighted additional differences between stMI-01 and stMI-03, and the btMI-05 and btMI-09 preparations, but at the same time resulted in quite similar lesions, suggesting the potential existence of two highly similar strains. In any case, despite the apparent differences underscored by the

inoculation in distinct models, it is important to take into account that TgMoL108I to wild type mouse, wild-type to wild-type, and TgVole to TgVole transmissions have been considered together to highlight differential strain features. Both sequential transmission barriers and dissimilar strain competition could have influenced the results, making strain typing of recombinant preparations a complex issue that precludes robust conclusion on the final number of distinct strains. Differences between rodent models upon inoculation of the same prion strain have been previously reported [75]. However, in our case, the fact that some preparations could show similarity in some models while differences in others, together with the variability found in a single group of animals, especially for stMI-01 and stMI-03, suggest that the phenotypic differences are independent of host-exclusive factors and more likely related to conformer mixtures and their relative stability. Nonetheless, since no more than two serial transmission experiments were performed in the different animal models for each PMSA product, we cannot exclude the possibility that they are not fully adapted or stabilized to the brain environment, requiring additional passages for that purpose. This fact could also contribute to the heterogeneity of the neuropathological lesions found within some groups and could also have an impact on the final lesion profiles characteristic of each preparation, that could be slightly different from those of the fully adapted strains. Thus, considering cautiously the results from the second passages and attending to the distinct rodent prion strain classes based on their stability upon transmission *in vivo* [74], btMI-05 and especially btMI-09 are reminiscent of a Class II strain, highly stable in the same host, altered when inoculated in a different host, but able to stabilize rapidly in the new environment. In contrast, stMI-01 could be related to a class IV strain, non-adaptive prions unable to adapt in this case to the wild-type mouse PrP. However, results in TgVole suggest a minor component able to propagate efficiently in this model, likely the same conformer arising from the adaptation of btMI-05 and btMI-09 to this model, which could be the most thermodynamically favorable conformation. Finally, stMI-03, given its unusual behavior in terms of incubation period upon transmission to wild-type mice (accelerated despite the existing polymorphic barrier), could be associated with a Class III strain, with a conformer favorable to wild-type mice becoming the most abundant in a single passage. Overall, the PMSA products seem to be composed of different and heterogeneous conformer mixtures, probably featuring higher variability than brain-derived prions. This is possibly due to the lack of restrictions imposed by glycosylation, GPI-anchoring, or complex interactions with host factors. Differences in the exact composition of each mixture could explain the variations observed for each preparation when inoculated in the distinct models. However, all show a clear trend towards the emergence or selection of the most favored conformer, as indicated by the convergence of all preparations upon secondary transmissions in wild-type mice and TgVole.

Changing the propagation environment beyond the PrP can also lead to prion strain evolution or selection, as previously demonstrated using cell culture media complemented with anti-prion agents [73]. In our case, eliminating dextran sulfate from the propagation media seems to exert an equivalent effect on stMI-03 and btMI-09 PMSA products. Through the lens of strain classes according to their stability upon propagation [74], the dramatic change in biochemical and biological features of stMI-03 CB further supports its behavior as a Class III. However, in this case, the absence of cofactor could impede the breakdown of the conformer favorable to wild-type mouse PrP, whereas conformers able to propagate in TgVole emerge with some phenotypic differences compared to the original dextran-complemented product. In contrast, btMI-09 propagated sequentially in dextran-free and dextran-complemented substrates, despite showing distinct PrP$^{res}$ electrophoretic mobility profiles, does not suffer any notable alteration of the disease phenotype when inoculated in TgMoL108I. This is expected for a Class II strain, which furthermore seems identical in all aspects when propagated back

in the presence of dextran. Nonetheless, some alteration could be noticed upon inoculation of btMI-09 dex and btMI-09 CB preparations in TgVole, where only dextran-complemented preparations seem to target the striatum. This indicates a potential change in strain properties, although, alternative explanations cannot be ruled out without additional passages in this model. The changes in electrophoretic mobility patterns of both PMSA products after adaptation to a cofactor-free environment raised questions about the relationship between specific fragments and the infectivity or strain properties of different recombinant prions. Initially, the presence of an approximately 16 kDa band, corresponding to the amyloid core after N-terminal digestion in brain-derived prions [76], was considered essential for *in vivo* infectivity. This band was used as a selection criterion to confirm rec-PrP misfolding in dextran sulfate-supplemented PMSA reactions [44], as our previous work with bank vole rec-PrP strongly suggested it indicated *bona fide* prion misfolding [45]. Nonetheless, the infectivity of btMI-09 adapted to CB despite lacking this band suggests that structurally distinct misfolded rec-PrP conformers can act as *bona fide* prions in cofactor-free conditions. This aligns with the diverse PIRIBS structures found in scrapie-derived rodent prions [4,6] and those causing Gerstmann-Sträussler-Scheinker syndrome with F198S mutation [77]. Albeit different post-PK electrophoretic profiles typically indicate distinct structural arrangements based on protease accessibility, the similar behavior in TgMoL108I of btMI-09 propagated in the presence and in the absence of dextran suggests other factors, including cofactor binding, may influence prion electrophoretic profiles. This was confirmed when dextran sulfate prevented complete digestion of the 16 kDa fragment in btMI-09 CB and CB2 preparations, consistent with recent structural studies showing electrodense particles (potentially polyanionic cofactors) strongly associated with prion fibrils [4–6,8,9]. Alternatively, the adaptation to different environments might have induced subtle conformational changes that significantly affected electrophoretic mobility but minimally impacted pathobiological or strain features, suggesting that distinct conformers could show highly resemblant pathobiological characteristics despite apparent structural differences. All in all, the presence or absence of dextran sulfate in the propagation environment appears to strongly influence the pathobiological features of stMI-03, while presenting a milder or possibly negligible effect on the btMI-09 preparation. Since the description of RNA as a promoter of prion propagation *in vitro* [36], the role of polyanionic cofactors in prion misfolding has been studied both *in vitro* and *in vivo*. The fact that shortening heparan sulfate chains [61] or eliminating sulfation of heparan sulfate [78] prolongs survival in prion-infected animals clearly demonstrated an important role also *in vivo*. However, it was unclear whether they act solely as promoters of prion propagation or could play some other role on prion formation, infectivity, brain area tropism, or determination of the pathobiological features of prion strains. Deleault and colleagues showed that RNA molecules could also stimulate spontaneous *bona fide* prion misfolding *in vitro* [29]. This, together with the observation of selective incorporation of RNA molecules of specific sizes into hamster PrP fibrils [79] and the adaptation of prions to cofactor-free environment *in vitro* [56,80], led them to propose that cofactors were integral and necessary components of infectious prions. Additionally, they suggested that cofactors can restrict strain properties, acting as selectors of specific strains and thus defining their neurotropism [37]. The results shown here also support a misfolding-facilitating role of the dextran sulfate when used as a cofactor for *in vitro* prion generation. The combination of the isoleucine at position 108 and the presence of dextran sulfate allows obtaining spontaneously misfolded and highly infectious prions in under 24 h. In addition, and in agreement with our previous research [39], the cofactor seems to play an important role in defining and maintaining the pathobiological features of distinct prion strains, in line with the "cofactor selection" model [37]. Nonetheless, our results prove that infectious prions (named here as MoL108I-CB-01 and MoL108I-CB-02)

can be obtained spontaneously in absence of the polyanionic cofactor dextran sulfate. This clearly indicates that polyanionic cofactors are not necessary for the formation of infectious prions, and thus, cannot be considered an integral component of all prions, according to the potential strain dependent effect observed here. Additionally, our results from the sequential propagation experiments performed with stMI-03 and btMI-09 preparations indicate that the same cofactor can show different influences on potentially distinct conformers, in agreement with previous research [81]. This is evident as propagation in the absence of cofactor leads to dramatic changes in one case while showing a very limited effect, if any, on the characteristics of btMI-09, only noticeable upon inoculation in TgVole. Differential interaction of heparin with PrP$^C$ from different species has already been shown [82]. However, taking into account the same sequence of our preparations, the most probable explanation lies in the differential interactions of the majoritarian conformers in each preparation. Along this line, previous results in which initially infectious prions generated in the presence of cofactors lost their infectivity upon propagation in a cofactor-free environment [56] could be reconciled with our results through a phenomenon similar to that observed here for stMI-03. Potentially, a Class II or Class III recombinant prion could have been obtained, highly dependent on the presence of the cofactor to maintain its pathobiological features and prone to evolve (be it through selection or adaptation) upon propagation in different environments. This could give rise to apparently non-infectious conformers, such as our stMI-03 CB, that could readily recover the original properties upon propagation in a brain homogenate environment.

## Conclusions

Altogether, we have demonstrated that cofactors are not necessary to generate infectious, cross-species transmissible recombinant prions *in vitro*. However, polyanionic cofactors such as the dextran sulfate used here are able to stimulate this event and limit or restrict the diversity of the conformers that can be obtained. Furthermore, potential conformer mixtures could have been formed, likely featuring distinct stability and alteration capacity and showing variable dependence on the presence of the cofactor to maintain their pathobiological or strain features. These findings contribute to our knowledge on the spontaneous misfolding of prions, the main event underlying sporadic or idiopathic prion diseases, the most frequent among these devastating neurodegenerative disorders. Our results indicate that misfolded PrP or PrP$^{Sc}$-independent factors, such as polyanionic cofactors, are unnecessary for prions to form spontaneously and acquire the ability to cause a transmissible spongiform encephalopathy. However, due to their capacity to stimulate such phenomenon *in vitro*, and considering their influence on the pathobiological characteristics of the PrP$^{Sc}$, they could be highly relevant on triggering PrP$^C$ misfolding and on defining the prion strain variety observed in humans and other mammals suffering from prion diseases.

## Supporting information

**S1 Table.  List of forward (Fw) and reverse (Rv) primers used for the generation of the 20 mouse rec-PrP variants with all possible amino acids in position 108.** The mutated codon is highlighted in bold in all primers used for site-directed mutagenesis. Primers used for the 5' and 3' extremes in all cases are shown under the name L108 (wild-type), with the nucleotides belonging to the PrP ORF underlined.
(PDF)

**S1 Fig.  Electrophoresis and total protein staining of the 20 PMSA substrates prepared with mouse PrP variants with all naturally occurring amino acids at position 108.** Substrates were

evaluated to confirm comparable rec-PrP concentrations prior to assessing their spontaneous misfolding capacity. Except for the L108C variant, which formed dimers (indicated by an asterisk) due to potential disulfide bridge formation, all other variants exhibited similar concentrations and were deemed suitable for PMSA. Despite efforts to concentrate the L108C substrate, dimer formation limited its preparation. The assay proceeded with the available substrate, acknowledging that while the lower concentration might underestimate misfolding propensity, enhanced misfolding would remain detectable as misfolding is not significantly affected within a certain concentration range [1]. MW: Molecular weight marker.
(PDF)

**S2 Fig. A) Proteinase K resistance assay of PMSA products (stMI-01, stMI-03, btMI-05, btMI-09).** The relative resistance to PK digestion of the four PMSA products was evaluated to determine their prion-like characteristics and assess potential conformational differences. Products were digested with increasing PK concentrations (25–2000 μg/ml) at 42 °C for 45 minutes, followed by electrophoresis and protein staining. Additionally, three independent digestion reactions were performed and rec-PrP$^{res}$ levels evaluated through Western blot and densitometric analysis, the results of which are plotted and normalized using the samples digested with 25 μg/ml of PK as reference of maximum, 100%, signal intensity. Normalized data from the densitometric analysis, revealed similar resistance for stMI-01, btMI-05, and btMI-09, with no significant differences. In contrast, stMI-03 exhibited significantly lower resistance compared to stMI-01 (p = 0.013), btMI-05 (p = 0.013), and btMI-09 (p = 0.028), indicating distinct conformational properties. An undigested substrate control (rec-MoL108I) was included for size reference. MW: Molecular weight marker. **B) Evaluation of the self-propagation capacity of the misfolded recombinant PMSA products *in vitro* in homologous and C) heterologous recombinant PrP-containing substrates.** The ability of the PMSA products to induce misfolding of homologous mouse L108I recombinant PrP was tested via serial dilutions ($10^{-1}$ to $10^{-11}$) in PMSA. All products propagated efficiently up to at least a $10^{-8}$ dilution, with btMI-09 reaching $10^{-11}$. PK-resistant misfolded rec-PrP was detected via digestion, electrophoresis, and total protein staining, with results displayed in shades of grey to indicate the percentage of positive replicates. For the evaluation of propagation capacity in a heterologous substrate, PMSA products were tested in mouse wild-type L108 recombinant PrP substrate ($10^{-1}$ dilution, three replicates). Most seeds propagated in this heterologous substrate, confirming their prion-like properties. However, stMI-03 failed to propagate under these conditions, likely due to strain-specific polymorphic barriers consistent with its differential PK resistance. PK: Proteinase K; MW: Molecular weight marker.
(PDF)

**S3 Fig. Negative staining electron microscopy micrographs of the four PMSA products selected for further characterization.** The images display sixteen representative micrographs for each of the PMSA products **A)** stMI-01, **B)** stMI-03, **C)** btMI-05, and **D)** btMI-09 after partial purification through ultracentrifugation in a density gradient. Partially purified samples were stained with uracil acetate and imaged with a transmission electron microscope JEM-1230 (JEOL) at 100 kV, equipped with a CCD Orius SC1000 (GATAN) camera. All four PMSA products show fibrillar structures reminiscent of brain-derived prion rods, with virtually indistinguishable ultrastructure. In all four cases, clusters of yet unidentified electrodense material (Em) were observed near the rods. Whether these are a contaminant from the purification process or biologically significant for fiber formation is yet to be determined. Notably, in the btMI-09 preparation (panel D), apart from the major fiber type found in the other preparations, two additional fibril populations were observed. The micrographs revealed some thinner fibers (Tf, thin fibers) and clusters of especially short rods (Sr, short rods) that were

not observed in the other PMSA products. **E)** The most notable structural findings or shared elements among the four PMSA preparations have been highlighted in a selection of micrographs. In most fibers, two parallel axial densities of approximately 12 nm were observed, resembling rail tracks. In all preparations, straight (S, in orange) and curved (C, in yellow) fibers were detected, which could represent distinct fiber populations. Lateral clustering of fibers organized in bundles was another common finding in all the products analyzed. The presence of torsions (indicated by yellow arrows) in some fibers suggests a helicoidal symmetry, although most seemed flat or presented long half-pitches that precluded their identification. Each image in the group of sixteen contains a link that opens a higher resolution version in a web browser when clicked.
(PDF)

**S4 Fig. Assessment of the capacity of the four selected recombinant misfolded PrP generated by PMSA to induce misfolding of PrP$^C$ from brain *in vitro*.** To predict the potential infectivity *in vivo* of the distinct PMSA products selected for further characterization (stMI-01 Dx, stMI-03 Dx, btMI-05 Dx and btMI-09 Dx), their capacity to induce misfolding of PrP$^C$ in brain homogenates of TgMoL108I animals, expressing 3-fold the mouse L108I PrP, was evaluated using PMCA. PMSA products were partially purified by ultracentrifugation through a density gradient, resulting in visible halos of proteic aggregates in all four samples. These purified fractions exhibited indistinguishable biochemical properties after proteinase K digestion and retained the same electrophoretic pattern as the original product. These purified fractions were used to seed a PMCA substrate based on TgMoL108I brain homogenate at 1:10 dilutions, and a 24 h PMCA reaction was performed (R1). Two additional serial PMCA rounds were conducted, with the second round (R2) seeded at 1:10 dilution using the product from the first round, and the third round (R3) utilizing a 1:10 dilution of the product form the second one. After the three serial PMCA rounds of 24 h, PrP$^{Sc}$ detection was carried out by proteinase K digestion and Western blotting (mAb Sha31 at a dilution of 1:4,000). A seeded tube at 1:10 but not submitted to PMCA is also included for each PMSA product, referred to as R0, to show the signal corresponding to the recombinant seed, slightly lower than the unglycosylated PrP$^C$ from brain. Finally, an unseeded tube was also included together with each PMSA product in every PMCA round, performing also serial passages as control for cross-contamination or spontaneous misfolding. All four recombinant seeds were able to misfold brain-derived PrP$^C$ from the first PMCA round, giving rise to the classical three-banded PrP$^{Sc}$ pattern, suggesting the potential infectivity of these preparations *in vivo*. PK: Proteinase K; NBH control: Normal brain homogenate from TgMoL108I. Mw: Molecular weight.
(PDF)

**S5 Fig. Anatomopathological analysis of C57BL/6 mice inoculated intracerebrally with brain homogenate from a spontaneously sick TgMoL108I transgenic mouse.** Histopathological assessment of spongiform lesions and PrP$^{res}$ deposits of C57BL/6 mice after serial inoculation (1st and 2nd passages) with brain homogenate from a terminal ill TgMoL108I mouse with spontaneous disease. Analysis revealed no spongiform changes upon hematoxylin and eosin staining (H&E), as shown in the thalamic region of representative animals. PrP$^{res}$ deposits were undetectable, using 6C2 mAb (1:1,000), demonstrating that this spontaneously generated prion strain failed to transmit to wild type mice. H&E: Hematoxylin and eosin; IHC: Immunohistochemistry.
(PDF)

**S6 Fig. Brain lesion and PrP$^{res}$ deposit distribution of the wild-type mice-passaged recombinant PMSA products after secondary transmission in wild-type mice.** Histopathological

assessment of spongiform lesions and PrP^res deposits of PMSA preparations stMI-03, btMI-05 and btMI-09 after secondary transmission into C57BL/6 mice(which was unsuccessful already at first passage for stMI-01) shows moderate to intense spongiform changes upon hematoxylin and eosin staining (H&E), as can be seen in the thalamic region of representative animals from each group. PrP^res deposits, labeled with 6C2 mAb (1:1,000), were detectable but very faint, all showing punctate or granular deposits associated with glia or neurons and in the neuropil (see digitally enlarged images on the left). Spongiform lesion profiles and PrP^res deposition profiles, shown on the right, represent the mean semi-quantitative scoring (0–4, vertical axis, ± standard error of the mean -error bars-) of the spongiform lesions (continuous line, black) and the immunohistochemical labelling of PrP^res deposits (dashed line, black) against 14 brain regions. Although there are some differences, mostly in terms of spongiform lesion intensity, stMI-03, btMI-05, and btMI-09-inoculated animals show highly coincident lesions and PrP^res staining, suggesting infection by the same strain or the convergence of the recombinant preparations *in vivo* to the same strain through selection or adaptation. H&E: Hematoxylin and eosin staining; IHC: Immunohistochemistry.
(DOCX)

**S7 Fig. Biochemical analysis and survival curves of TgVole (1x) mice inoculated with spontaneously misfolded PMSA products. A) Biochemical analysis of TgVole (1x) brains inoculated with four selected PMSA products**. Brain homogenates from all inoculated animals showing clinical signs of transmissible spongiform encephalopathy were analyzed by proteinase K (PK) digestion, electrophoresis, and Western blot (Sha31, 1:4,000). Results revealed the presence of classical three-banded pattern PrP^Sc, demonstrating the infectious capacity and *bona fide* nature of the recombinant prions generated spontaneously by PMSA and their cross-species transmissibility. The gel shows two representative samples from each group inoculated with the distinct recombinant products and one of the brain-derived RML and 22L prions, also passaged in TgVole (1x). All the samples are indistinguishable from each other except for those from stMI-03 inoculated animals, among which two electrophoretic patterns could be found, one of them showing a lower unglycolsylated band. **B) Kaplan-Meier survival curves of TgVole (1x) inoculated with PMSA products and classical murine prion strains**. Kaplan-Meier survival curves illustrate the incubation periods following intracerebral inoculations with the distinct PMSA products, RML, and 22L. stMI-01 shows great dispersion in incubation periods, while btMI-05 and btMI-09 exhibit profiles more similar to brain-derived strains RML and 22L, with lower dispersion but in the case of btMI-09 with a shorter incubation period, indicating its' high cross-species transmissibility. Groups showing greater dispersion might indicate stronger barriers due to strain characteristics, mixtures of conformers undergoing slightly different selection process in each animal, or the formation of unstable or somewhat immature conformers, although likely due to the high susceptibility of bank vole PrP^C to misfolding, most of the inocula show much lower dispersion than for previous models. PK: Proteinase K; NBH: Undigested normal brain homogenate; MW: Molecular weight marker.
(PDF)

**S8 Fig. Brain lesion and PrP^res deposit distribution of the TgVole (1x)-passaged recombinant PMSA products after secondary transmission in the same model.** Histopathological assessment of spongiform lesions and PrP^res deposits of TgVole (1x)-passaged PMSA preparations stMI-01, stMI-03, btMI-05, and btMI-09 after secondary transmission in TgVole (1x) shows clear spongiform changes upon hematoxylin and eosin staining (H&E). Micrographs of the temporal cortex and striatum of representative animals from each group are shown, notice intense spongiosis in the striatum in all groups, but only involvement of temporal cortex in

the case of btMI-05. For stMI-01 and stMI-03-inoculated animals, in some animals from each group, a few small plaques in the hippocampus and granular-coalescent deposits in the striatum were detected (indicated by black arrowheads) but the phenotype was not homogeneous among the group. In contrast, btMI-05 and btMI-09-inoculated animals showed distinct patterns homogeneously observed in all animals within each group: btMI-05: granular to coalescent labeling associated with striatum vacuoles and intense plaque labeling in the hippocampus. btMI-09: fine punctate intraneuronal patterns with plaques mainly in the hippocampus (indicated by black arrowheads). Spongiform lesion profiles and PrP^res deposition profiles, shown on the right, represent the mean semi-quantitative scoring (0–4, vertical axis, ± standard error of the mean -error bars) of the spongiform lesions (continuous line, black) and the immunohistochemical labeling of PrP^res deposits (dashed line, black) across 14 brain regions (Pfc: piriform cortex, H: hippocampus, Oc: occipital cortex, Tc: temporal cortex, Pc: parietal cortex, Fc: frontal cortex, cc: corpus callosum; S: striatum, T: thalamus, HT: hypothalamus, M: mesencephalon, Mob: medulla oblongata, Cm: cerebellar nuclei, Cv: cerebellar vermis, Cc: cerebellar cortex). Overall, despite subtle differences detected between groups that could suggest that the four recombinant preparations might represent distinct strains, yet the presence of the most prominent lesions (such as focalized spongiosis in striatum or plaques in the hippocampus) in some animals across all groups may also argue in favor of a single or highly similar strain. H&E: Hematoxylin and eosin staining; IHC: Immunohistochemistry. (PDF)

**S9 Fig. Biochemical analysis of TgMoL108I, wild-type and TgVole (1x) mice brains inoculated with spontaneously misfolded PMSA products.** Brain homogenates from all inoculated animals showing clinical signs of transmissible spongiform encephalopathy were analyzed by proteinase K (PK) digestion, electrophoresis, and Western blot (Sha31, 1:4,000). Results revealed the presence of classical three-banded pattern PrP^Sc, demonstrating the infectious capacity and *bona fide* nature of the recombinant prions generated spontaneously by PMSA and their cross-species transmissibility. The gel shows two representative samples from each group inoculated with the distinct recombinant products and one of the brain-derived RML as control of a classical prion strain. PrP^Sc from all three models inoculated with btMI-09 CB are indistinguishable from each other except for the amount of PrP^Sc, higher in wild-type likely due to their long incubation period. stMI-03 inoculated TgVole (1x) animals, are characterized by a lower molecular weight unglycosylated band, that could correspond to the selection of one of the two distinct conformers hinted previously for stMI-03 dex upon inoculation in TgMoL108I. PK: Proteinase K; NBH: Undigested normal brain homogenate; MW: Molecular weight marker.
(PDF)

**S10 Fig. Brain lesion and PrP^res deposition induced by recombinant prions stMI-03 and btMI-09 propagated in cofactor-devoid environment (CB) were compared to their original dextran sulfate-supplemented preparations across three animal models (TgMoL108I, wild-type, and TgVole (1x) mice). A)** Spongiform lesion (continuous line, black) and PrP^res deposition (dashed line, grey) profiles were scored semi-quantitatively (0–4) across 14 brain regions, revealing distinct transmission patterns. stMI-03 CB successfully induced disease only in TgVole (1x) mice, showing mild and localized striatal spongiform lesions that differed from the more severe lesions caused by stMI-03 dex. Conversely, btMI-09 CB induced disease in all models with variations in lesion profiles. In TgMoL108I, btMI-09 CB caused milder hippocampal but more severe cerebellar lesions compared to btMI-09 dex, with consistent PrP^res deposits patterns. In wild-type mice, btMI-09 CB induced strong spongiform lesions in brainstem and cortices with minimal PrP^res labeling. In TgVole (1x), btMI-09 CB caused mild

lesions with intraneuronal PrP$^{res}$ deposits in brainstem, cerebellar cortex, and hippocampus. **B)** A more detailed comparison of the lesions from wild-type mice inoculated with btMI-09 dex and btMI-09 CB illustrates the slightly higher severity of spongiform lesions in the latter through hematoxylin and eosin staining (H&E) of the thalamic region, and the similarity of the mild granular PrP$^{res}$ deposit pattern (see digitally enhanced sections on the left), labeled with 6C2 (1:1,000). Images of the btMI-09 dex-inoculated mice are the same as those used in S5 Fig to facilitate comparison. When plotted together, the spongiform lesion profile similarity between btMI-09 dex (red line) and btMI-09 CB (black line) is evident and suggests conservation of the pathobiological features of the original preparation in the cofactor-devoid version. **C)** Conversely, in TgVole (1x) mice inoculated with btMI-09 CB and btMI-09 dex shows the main differences observed in this model, namely the severe and focalized spongiform lesions, evident upon H&E staining, and PrP$^{res}$ plaques, labeled with 6C2 (1:1,000), in the striatum of the dextran-complemented preparation, which are absent in btMI-09 CB-inoculated animals. Images of the btMI-09 dex-inoculated TgVole (1x) mice are the same as those used in S7 Fig to facilitate comparison. In this case, when plotted together, the spongiform lesion profile (± standard error of the mean -error bars-) clearly displays the differences found for the two preparations (btMI-09 dex in red and btMI-09 CB in black) in this model. H&E: Hematoxylin and eosin staining; IHC: Immunohistochemistry.
(PDF)

**S11 Fig. Brain lesion and PrP$^{res}$ deposit distribution in TgMoL108I mice inoculated with btMI-09 prion sequentially adapted across cofactor-devoid and dextran sulfate-complemented PMSA substrates.** Histopathological assessment of spongiform lesions and PrP$^{res}$ deposits of TgMoL108I mice inoculated with the PMSA preparations btMI-09 dex, btMI-09 CB, btMI-09 dex$^2$, and btMI-09 CB$^2$. These preparations are products of the sequential adaptation of the original btMI-09 dex recombinant prion in cofactor-devoid and dextran sulfate-complemented PMSA substrates. The assessment aimed to compare the pathobiological features of all four preparations. **A)** Hematoxylin and eosin staining (H&E) shows moderate spongiform lesions in the thalamus and absence of spongiosis in the cerebellar cortex. The pattern of PrP$^{res}$ deposits, labeled with 2G11 monoclonal antibody (1:100), reveals PrP$^{res}$ small aggregates in the thalamus (indicated by black arrowheads) and larger plaques, characteristically located in the white matter of the cerebellar cortex. Spongiosis distribution profiles and PrP$^{res}$ deposition profiles for each group, shown below, represent the mean semi-quantitative scoring (0–4, vertical axis, ± standard error of the mean -error bars-) of the spongiform lesions (continuous line, black) and the immunohistochemical labeling of PrP$^{res}$ deposits (dashed line, black) across 14 brain regions. **B)** Spongiform lesion and PrP$^{res}$ deposit profiles plotted together (btMI-09 dex in yellow, btMI-09 CB in grey, btMI-09 dex$^2$ in black, and btMI-09 CB$^2$ in black) illustrate the high similarity of localization and intensity of the spongiform lesions and PrP$^{res}$ deposits in all cases. This suggests no significant alteration of the pathobiological features of the original btMI-09 preparation in this model occur related to the presence or absence of the cofactor in the propagation environment. H&E: Hematoxylin and eosin staining; IHC: Immunohistochemistry.
(PDF)

**S12 Fig. Negative staining electron microscopy micrographs of PMSA products stMI-03 and btMI-09 propagated in cofactor-free substrate, and btMI-09 back-passaged to dextran sulfate-complemented substrate and again to a cofactor-free environment.** The images display four representative micrographs for each of the following PMSA products: stMI-03 dex (original preparation), stMI-03 CB, adapted to a cofactor-free substrate, btMI-09 dex (original preparation), btMI-09 CB (adapted to a cofactor-free substrate), btMI-09 dex$^2$ (resulting from

the back-passage of btMI-09 CB to a dextran-complemented substrate), and btMI-09 CB$^2$ (the product of readapting btMI-09 dex$^2$ to a substrate without cofactor). Samples were partially purified through ultracentrifugation in a density gradient, stained with uranyl acetate, and imaged with a transmission electron microscope JEM-1230 (JEOL) at 100 kV, equipped with a CCD Orius SC1000 (GATAN) camera. All PMSA products show fibrillar structures reminiscent of brain-derived prion rods, with no notable differences from the original dextran-complemented products that could explain the observed changes in biochemical and biological features. In all cases, the most remarkable structural features previously observed are conserved upon propagation in different substrates, including rods with two parallel axial densities, presence of straight and curved fibers, unidentified electrodense material, and high propensity for lateral clustering. Each image in the group of four contains a link that opens a higher resolution version in a web browser when clicked.
(PDF)

**S13 Fig. Biochemical analysis of secondary transmissions of spontaneously misfolded PMSA products to wild-type mice.** Brain homogenates from C57BL/6 mice inoculated in second passage with the recombinant PMSA products after successful infection of either TgMoL108I or wild type mice in first passage were analyzed by proteinase K (PK) digestion, electrophoresis, and Western blot (Sha31, 1:4,000). Results revealed the presence of indistinguishable classical three-banded PrP$^{Sc}$ patterns, characterized by the prominence of the diglycosylated band contrasting with control 22L and RML prions transmitted in the same models. The gel shows one representative sample from each group inoculated with different recombinant products, alongside brain-derived RML and 22L strains. PK: Proteinase K; NBH: Undigested normal brain homogenate; MW: Molecular weight marker.
(PDF)

**S14 Fig. Negative staining electron microscopy micrographs of two PMSA products generated spontaneously in the absence of cofactor.** The images display sixteen representative micrographs for each of the PMSA products MoL108I-CB-01 and MoL108I-CB-02 after partial purification through ultracentrifugation in a density gradient. Partially purified samples were stained with uranyl acetate and imaged with a transmission electron microscope JEM-1230 (JEOL) at 100 kV, equipped with a CCD Orius SC1000 (GATAN) camera. The two PMSA products show fibrillar structures reminiscent of brain-derived prion rods, with indistinguishable ultrastructural features. As in all previous preparations characterized by TEM, clusters of unidentified electrodense material (Em) were observed near the rods. Whether these are contaminants from the purification process or biologically significant for fiber formation is yet to be determined. Despite their high similarity to previous dextran-complemented preparations, with many fibers showing two parallel axial densities of approximately 12 nm resembling rail tracks (parallel yellow lines), lateral clustering of fibers and arrangement in the form of bundles is apparently lower (pointed by yellow lines), as is the presence of curved fibers (C, in yellow). In contrast, the presence of torsions (indicated by yellow arrows) or twisted fibers was higher than in previous preparations. Each image in the group of sixteen contains a link that opens a higher resolution version in a web browser when clicked.
(PDF)

## Acknowledgements

The authors would like to thank the following for their support: IKERBasque foundation, vivarium, maintenance and Electron Microscopy Platform from CIC bioGUNE for outstanding assistance; Maria de la Sierra Espinar and the rest of the personnel from CReSA for the

technical support; and Neiker and CEBEGA's biocontainment unit staff for excellent care and maintenance of the animals. The authors would also like to acknowledge the work from past laboratory members of the Prion Research Lab from CIC bioGUNE, that despite not directly involved in the manuscript have contributed along the years to the development of all the methods and techniques currently used in the laboratory. **Use of a generative AI tool to generate the clipart of a mice:**

A drawing of a mouse is used in Figs 6 and 7 of this manuscript to illustrate the intracerebral inoculations performed in distinct mouse models. This drawing or clipart was generated using the Copilot Designer generative AI tool (Microsoft365). The prompt introduced to get the image was as follows: "I need a realistic cartoon of a mouse in black and white or greyscale to illustrate a figure in a scientific paper. The image should clearly allow identifying the species, but do not try to mimic a real mouse, it should be like drawn by hand".

## Author contributions

**Conceptualization:** Miguel A. Pérez-Castro, Hasier Eraña, Jesús R. Requena, Joaquín Castilla.

**Formal analysis:** Enric Vidal, Jorge M. Charco, Nuno Gonçalves-Anjo, Samanta Giler.

**Funding acquisition:** Manuel A. Sánchez-Martín, Jesús R. Requena, Joaquín Castilla.

**Investigation:** Miguel A. Pérez-Castro, Hasier Eraña, Enric Vidal, Jorge M. Charco, Nuria L. Lorenzo, Nuno Gonçalves-Anjo, Josu Galarza-Ahumada, Carlos M. Díaz-Domínguez, Patricia Piñeiro, Ezequiel González-Miranda, Samanta Giler, Manuel A. Sánchez-Martín, Mariví Geijo, Jesús R. Requena.

**Methodology:** Miguel A. Pérez-Castro, Enric Vidal, Jorge M. Charco, Nuria L. Lorenzo, Carlos M. Díaz-Domínguez, Patricia Piñeiro, Glenn Telling, Manuel A. Sánchez-Martín, Joseba Garrido, Mariví Geijo.

**Project administration:** Miguel A. Pérez-Castro, Joaquín Castilla.

**Resources:** Enric Vidal, Glenn Telling.

**Supervision:** Hasier Eraña, Joseba Garrido, Mariví Geijo, Jesús R. Requena, Joaquín Castilla.

**Validation:** Nuno Gonçalves-Anjo, Carlos M. Díaz-Domínguez, Ezequiel González-Miranda.

**Visualization:** Josu Galarza-Ahumada, Patricia Piñeiro.

**Writing – original draft:** Miguel A. Pérez-Castro, Hasier Eraña, Joaquín Castilla.

**Writing – review & editing:** Miguel A. Pérez-Castro, Hasier Eraña, Enric Vidal, Jesús R. Requena, Joaquín Castilla.

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
