## [Decision Letter · Decision Letter 0]

4 Nov 2024

PPATHOGENS-D-24-02149Cofactors facilitate bona fide prion misfolding in vitro but are not necessary for the infectivity of recombinant murine prionsPLOS Pathogens Dear Dr. Castilla, Thank you for submitting your manuscript to PLOS Pathogens. After careful consideration, we feel that it has merit but does not fully meet PLOS Pathogens's publication criteria as it currently stands. Therefore, we invite you to submit a revised version of the manuscript that addresses the points raised during the review process. Please submit your revised manuscript within 60 days Jan 03 2025 11:59PM. If you will need more time than this to complete your revisions, please reply to this message or contact the journal office at plospathogens@plos.org. Please include the following items when submitting your revised manuscript:* A rebuttal letter that responds to each point raised by the editor and reviewer(s). You should upload this letter as a separate file labeled 'Response to Reviewers '. This file does not need to include responses to any formatting updates and technical items listed in the 'Journal Requirements' section below.* A marked-up copy of your manuscript that highlights changes made to the original version. You should upload this as a separate file labeled 'Revised Manuscript with Track Changes '.* An unmarked version of your revised paper without tracked changes. You should upload this as a separate file labeled 'Manuscript '. If you would like to make changes to your financial disclosure, competing interests statement, or data availability statement, please make these updates within the submission form at the time of resubmission. Guidelines for resubmitting your figure files are available below the reviewer comments at the end of this letter. We look forward to receiving your revised manuscript. Kind regards, Amanda L. WoermanAcademic EditorPLOS Pathogens Neil MabbottSection EditorPLOS Pathogens Michael Malim

Editor-in-Chief

PLOS Pathogens

orcid.org/0000-0002-7699-2064   **Journal Requirements:** **Additional Editor Comments (if provided):****Reviewers' Comments:** Reviewer's Responses to Questions

**Part I - Summary**

Reviewer #1: The manuscript by Pérez-Castro et al. presents extensive data on the spontaneous generation of infectious prions from recombinant PrP via Protein Misfolding Shaking Amplification (PMSA) in different environments, including dextran sulfate as a cofactor. Their findings challenge previous assertions by demonstrating that cofactors are not essential for acquisition of infectivity or strain characteristics.

The study is comprehensive and integrates data effectively from multiple experimental models. However, given the substantial amount of data and to maintain clarity and ensure accessibility, the presentation could benefit from simplified / more descriptive legends to aid readability. Some sections, particularly those comparing the neuropathological properties of the recombinant prion strains, are particularly dense and could be streamlined.

To strengthen the interpretation of the data, further clarification on controls and statistical methods is necessary.

Reviewer #2: The manuscript by Pérez-Castro et al. describes novel synthetic prions generated by mutated mouse prion protein amplified in vitro and then passaged in vivo in various hosts. The conclusions drawn by this work is that generation of synthetic prions occurs without the need of cofactors.

Reviewer #3: Pérez-Castro and colleagues present an analysis of factors involved (or not) in allowing spontaneous in vitro formation of infectious prions. In an impressive array of in vitro prion propagation reactions and in vivo bioassays, they provide strong evidence that infectious prions can be formed spontaneously under very simple conditions in the absence of polyanionic cofactors. The work is quite thought-provoking and addresses fundamental issues of prion biology and biochemistry. It should be of real significance to the prion field. The presentation and interpretation of the results is thorough, nuanced and clear, for the most part (see below). However, I have some suggestions for improving the manuscript for the authors’ consideration.

**Part II – Major Issues: Key Experiments Required for Acceptance**

Reviewer #1: 1. Bioassays in different animal models.

- The study includes limited serial passages (one or two at most) within the same host model, i.e. without full terminal adaptation or stabilization of optimized conformers. This likely contributes to variability in neuropathological findings (e.g., lines 640-660 and elsewhere) and western blot analyses. Acknowledging this in the discussion would add context to the observed inconsistencies in strain stability.

- The tgMoL108I and tgVole 1x mouse models (I109) are not adequately referenced, leaving some uncertainty about the models’ documented behavior and the occurrence of spontaneous prion disease. Providing background on these models would increase confidence in the results.

For instance, the initial bioassay testing the infectivity of the stMI-01, stMI-03, btMI-05, and btMI-09 preparations in tgMoL108I mice mentions that these mice spontaneously accumulate prions with age, which are non-infectious to wild-type mice. This statement lacks a supporting reference, and inclusion of a control experiment using mock-inoculated tgMoL108I mouse brain in wild-type mice would clarify the claim and confirm any impact on the findings from the second passage in wild-type mice.

- Comparisons between direct transmission to wild-type mice and two-passage transmission via tgMoL108I mice suggest distinct strain properties. However, these different transmission paths introduce potential confounders. Sequential barriers and strain competition may influence conformer stability and adaptation, and acknowledging this complexity in the discussion would provide a more nuanced interpretation of the data.

- The authors note a significant reduction in incubation periods between the first and second passages when they transmitted stMI-01, stMI-03, btMI-05 and btMI-09 to wild-type mice, which they consider "extraordinary." However, reductions of this magnitude are common in prion studies. Additionally, to confirm stability and adaptation, a third passage may be necessary to observe full strain stabilization.

2. Biochemical analyses

- In comparing the PK resistance of PMSA-generated prions (stMI-01, stMI-03, btMI-05, and btMI-09), the data showing stMI-03’s increased PK resistance is not convincing, as the band at 2 mg/mL is barely visible, and the 1 mg/ml result is unclear due to apparent technical issue. Repeating these analyses with quantitative or statistical studies would strengthen this claim.

- Directly comparing on the same western blots the PrPres signatures after transmission with and without cofactor, including after PNGase treatment, would provide clearer insights into the structural stability of PMSA-generated prions under different conditions.

For example, Figure 5 displays PrPres profiles after direct transmission to wild-type mice, which show mostly a migration profile higher than with the RML or 22L strains. In tgI108M, the opposite is found. The PrPres signatures after one passage in tgI108M and another in wild-type mice are not shown, why?

- The role of various proteolytic fragments in PMSA-generated prions is not discussed. Given that the absence of dextran sulfate in PMCA experiments affects the presence of the 16 kDa fragment, discussing its potential contributions to infectivity or strain properties would add depth to the findings.

Reviewer #2: The work is well presented and executed. The manuscript itself is well-written and orderly organized.

While the methods are described in details and the experiments presented with a clear fashion, I only have a more fundamental scientific issue with this work.

Despite the wealth of citations employed in support of the presentation, discussion and conclusions of the work, fundamental previous work is overlooked. I hope that this problem is simple a forgetfulness issue by the authors but nevertheless essential for the readers that may not be accustomed with previous seminal work.

There are several papers that have shown that to obtain synthetic prions one does not need cofactors and these works are not cited in this manuscript.

The following papers several years ago have shown that it is possible to generate wild-type mouse synthetic prions without the use of sequence exchange of any amino acid. It has been described that there is no need to use bank vole sequence to propagate in a more permissive way any synthetic prion.

The papers are:

Synthetic mammalian prions.

Legname G, Baskakov IV, Nguyen HO, Riesner D, Cohen FE, DeArmond SJ, Prusiner SB. Science. 2004 Jul 30;305(5684):673-6. doi: 10.1126/science.1100195.

Strain-specified characteristics of mouse synthetic prions.

Legname G, Nguyen HO, Baskakov IV, Cohen FE, Dearmond SJ, Prusiner SB. Proc Natl Acad Sci U S A. 2005 Feb 8;102(6):2168-73. doi: 10.1073/pnas.0409079102. Epub 2005 Jan 25.

Continuum of prion protein structures enciphers a multitude of prion isolate-specified phenotypes.

Legname G, Nguyen HO, Peretz D, Cohen FE, DeArmond SJ, Prusiner SB. Proc Natl Acad Sci U S A. 2006 Dec 12;103(50):19105-10. doi: 10.1073/pnas.0608970103. Epub 2006 Dec 1.

Design and construction of diverse mammalian prion strains.

Colby DW, Giles K, Legname G, Wille H, Baskakov IV, DeArmond SJ, Prusiner SB. Proc Natl Acad Sci U S A. 2009 Dec 1;106(48):20417-22. doi: 10.1073/pnas.0910350106. Epub 2009 Nov 13.

Protease-sensitive synthetic prions.

Colby DW, Wain R, Baskakov IV, Legname G, Palmer CG, Nguyen HO, Lemus A, Cohen FE, DeArmond SJ, Prusiner SB. PLoS Pathog. 2010 Jan 22;6(1):e1000736. doi: 10.1371/journal.ppat.1000736.

Synthetic prions with novel strain-specified properties.

Moda F, Le TN, Aulić S, Bistaffa E, Campagnani I, Virgilio T, Indaco A, Palamara L, Andréoletti O, Tagliavini F, Legname G. PLoS Pathog. 2015 Dec 31;11(12):e1005354. doi: 10.1371/journal.ppat.1005354. eCollection 2015

In addition, what claimed to be a novel technique for generation of in vitro generated synthetic prions, PMSA, is instead a variation of techniques already in use for quite some time in several laboratories, namely amyloid seeding assay (ASA) and real time – quacking induced conversion (RT-QuIC).

The following papers describe these techniques for the first time several years ago:

Prion detection by an amyloid seeding assay.

Colby DW, Zhang Q, Wang S, Groth D, Legname G, Riesner D, Prusiner SB. Proc Natl Acad Sci U S A. 2007 Dec 26;104(52):20914-9. doi: 10.1073/pnas.0710152105. Epub 2007 Dec 20.

Ultrasensitive detection of scrapie prion protein using seeded conversion of recombinant prion protein.

Atarashi R, Moore RA, Sim VL, Hughson AG, Dorward DW, Onwubiko HA, Priola SA, Caughey B. Nat Methods. 2007 Aug;4(8):645-50. doi: 10.1038/nmeth1066. Epub 2007 Jul 22.

I would like to suggest that the authors cite these works accordingly prior to claim that they have found anything novel.

Again, the work is of interest and may deserve to be published but maybe a little biased toward something that is not completely novel.

Reviewer #3: 1) Lines 491-2: “…However, stMI-03 showed higher resistance up to 2000 μg/ml…” >>This is not apparent in the figure, especially comparing the 1000 ug/ml digestions. In fact, stMI-03 seems to be mostly digested at 1000 ug/ml then, paradoxically, is less digested at 2000. Also, the overall protein load seems to be a bit lower for stMI-03 than the others, confounding such comparisons. This experiment bears repeating, and perhaps, reinterpreting.

2) Bioassays: Perhaps I missed it, I was unable to find an accounting of the concentrations of PrP in these inocula, and how those amounts might compare to the concentration of PrPSc in the brain RML and 22L homogenates. This would allow consideration of the important issue of the relative infectivities per mass of PrP when considering the relative incubation periods. Even more informative in this regard would be bioassay data from dilutions of the inocula if available (e.g. if the inocula are diluted 10-fold or more, do the titers become sub-infectious? This would make them much less potent than the brain inocula. In any case, the issue of relative amounts of PrP in the PMSA vs brain inocula should be explained and discussed.

2) Lines 748-755: Regarding stMI-01 CB, btMI-05 CB, and btMI-09: It would be interesting/relevant to see the PK-resistant banding profiles of these strains if dextran sulfate were added back to these preparations just prior to PK digestion. Multiple lines of evidence (including cryo-EM structures -see refs listed below) suggest that DS and other polyanionic cofactors would bind to the aligned stacks of cationic sidechains in the N-lobe of the ordered core of prion fibrils and would likely block PK cleavage sites in that lobe. Thus, if any of these PMSA products maintained the essential features of the original assemblies formed in the presence of DS, then you might see a restoration of the original PK-resistance just by adding DS prior to PK treatment (analogous to what you saw with the Dex-containing PMSA rounds).

**Part III – Minor Issues: Editorial and Data Presentation Modifications**

Reviewer #1: 1. Ultrastructural data.

While transmission electron microscopy (TEM) data are presented, their relevance to the study’s primary conclusions is not entirely clear. Emphasizing only the key TEM results directly tied to the study’s main findings would streamline and strengthen the narrative.

2. Statistical Analyses:

A clearer discussion on consistency and variability in propagation assays would be beneficial. Statistical analyses (e.g., non-parametric tests to evaluate differences in incubation times, vacuolation profiles, or seeding activities) would provide a more robust comparison of the prion strains.

Reviewer #2: N/A

Reviewer #3: 1) Although I appreciate the thorough and nuanced explanations and interpretations, the manuscript is rather long and might become more digestible to readers if redundancies and other less vital verbiage were distilled down a bit.

2) Line 82: Add references to now-solved strain-dependent conformations? See why below.

3) Lines 222-223: Unclear. Is CB referring to a buffer or a substrate in a buffer? The composition of CB is critical in understanding appreciating the simplicity of the authors’ conditions. I would suggest describing it in on first mention in the Results, too. I had to spend a bit of time trying to track it down in the Methods.

4) Line 457: How large were PK-resistant products generated with the histidine, lysine and methionine-containing constructs?

5) Table 1 and line 587: I found it confusing for the attack rate for the 'no inoculum' group to be reported as 0/18 when they showed clinical disease as well as PrPres (even if the banding profile was unusual). It is important to know what proportion of these animals displayed these clinical signs as reflected in the stated average incubation period.

6) Line 1018: show should be showing?

7) Lines 1052-3 (and elsewhere): I would recommend that the authors consider these mechanisms in terms of the known cryo-EM structures of ex vivo prions, e.g. PMID: 34433091, PMID: 35831291, PMID: 36342968, PMID: 35831275, PMID: 36646960, and PMID: 39448454. In these papers one can see that the sidechains of residues analogous to bank vole residues 109 and 112, as well as 139 in the mouse- and hamster-adapted scrapie structures, interact closely to form tightly packed H-phobic cores that are likely to be important in stabilizing the overall fibril structure. Differences in the sidechain geometries of V, I, L and M residues likely influences how well they can pack (or not) to form this core, and how readily a mismatched PrPC molecule can add on to a preexisting template (or at least your data would seem consistent with that hypothesis). Also notable is the fact that residues 109 and 112 are near/within a cluster of basic residues with outwardly oriented sidechains that appear to interact with polyanionic cofactors (such as dextran sulfate) in a way that, as noted above, are likely to mask potential PK-cleavage sites.

8) Line 1068: Following from the previous comment, PMID: 36342968 is paper that is also particularly pertinent to the current manuscript because it compares prion strains produced in the same genotype of mouse (where the panel of potential cofactors are the same), unlike the case in ref 40 (as nice as ref 40 is).

9) Line 1142: “thus, cannot be considered an integral component of prions.” >>maybe this should be softened to say "cannot be considered an integral component of ALL prions'?

10) Line 1151: loss should be lost?

PLOS authors have the option to publish the peer review history of their article (what does this mean? ). If published, this will include your full peer review and any attached files.

**Do you want your identity to be public for this peer review?** For information about this choice, including consent withdrawal, please see our Privacy Policy .

Reviewer #1: No

Reviewer #2: No

Reviewer #3: **Yes: ** Byron Caughey

 **Figure resubmission:** While revising your submission, please upload your figure files to the Preflight Analysis and Conversion Engine (PACE) digital diagnostic tool, https://pacev2.apexcovantage.com/ . PACE helps ensure that figures meet PLOS requirements. To use PACE, you must first register as a user. Registration is free. Then, login and navigate to the UPLOAD tab, where you will find detailed instructions on how to use the tool. If you encounter any issues or have any questions when using PACE, please email PLOS at figures@plos.org. Please note that Supporting Information files do not need this step. If there are other versions of figure files still present in your submission file inventory at resubmission, please replace them with the PACE-processed versions. **Reproducibility:** To enhance the reproducibility of your results, we recommend that authors of applicable studies deposit laboratory protocols in protocols.io, where a protocol can be assigned its own identifier (DOI) such that it can be cited independently in the future. Additionally, PLOS ONE offers an option to publish peer-reviewed clinical study protocols. Read more information on sharing protocols at https://plos.org/protocols?utm_medium=editorial-email&utm_source=authorletters&utm_campaign=protocols

---

## [Decision Letter · Decision Letter 1]

6 Jan 2025

Dear Dr. Castilla,

We are pleased to inform you that your manuscript 'Cofactors facilitate bona fide prion misfolding in vitro but are not necessary for the infectivity of recombinant murine prions' has been provisionally accepted for publication in PLOS Pathogens.

Best regards,

Amanda L. Woerman

Academic Editor

PLOS Pathogens

Neil Mabbott

Section Editor

PLOS Pathogens

Sumita Bhaduri-McIntosh

Editor-in-Chief

PLOS Pathogens

orcid.org/0000-0003-2946-9497

Michael Malim

Editor-in-Chief

PLOS Pathogens

orcid.org/0000-0002-7699-2064

Reviewer Comments (if any, and for reference):

Reviewer's Responses to Questions

**Part I - Summary**

Reviewer #1: The authors have addressed all my concerns satisfactorily and modified the manuscript accordingly. I have no further comments. Congratulations on this excellent piece of work.

Reviewer #2: The authors make a case for the production of infectious synthetic prions without the need of cofactors.

Reviewer #3: The authors have made a good-faith effort to address my concerns and improved the manuscript.

**Part II – Major Issues: Key Experiments Required for Acceptance**

Reviewer #1: no further issues

Reviewer #2: The authors have addressed all my queries

Reviewer #3: None

**Part III – Minor Issues: Editorial and Data Presentation Modifications**

Reviewer #1: no further issues

Reviewer #2: N/A

Reviewer #3: none

PLOS authors have the option to publish the peer review history of their article (what does this mean? ). If published, this will include your full peer review and any attached files.

**Do you want your identity to be public for this peer review?** For information about this choice, including consent withdrawal, please see our Privacy Policy .

Reviewer #1: **Yes: ** Vincent Béringue

Reviewer #2: **Yes: ** Giuseppe Legname

Reviewer #3: **Yes: ** Byron Caughey

---

## [Editor Report · Acceptance letter]

Dear Dr. Castilla,

We are delighted to inform you that your manuscript, "Cofactors facilitate bona fide prion misfolding in vitro but are not necessary for the infectivity of recombinant murine prions," has been formally accepted for publication in PLOS Pathogens.

Best regards,

Sumita Bhaduri-McIntosh

Editor-in-Chief

PLOS Pathogens

orcid.org/0000-0003-2946-9497

Michael Malim

Editor-in-Chief

PLOS Pathogens

orcid.org/0000-0002-7699-2064